# A Neural Mean Embedding Approach for Back-door and Front-door Adjustment

**Liyuan Xu**
Gatsby Unit
liyuan.jo.19@ucl.ac.uk

**Arthur Gretton**
Gatsby Unit
arthur.gretton@gmail.com

## Abstract

We consider the estimation of average and counterfactual treatment effects, under two settings: *back-door adjustment* and *front-door adjustment*. The goal in both cases is to recover the treatment effect without having an access to a hidden confounder. This objective is attained by first estimating the conditional mean of the desired outcome variable given relevant covariates (the "first stage" regression), and then taking the (conditional) expectation of this function as a "second stage" procedure. We propose to compute these conditional expectations directly using a regression function to the learned input features of the first stage, thus avoiding the need for sampling or density estimation. All functions and features (and in particular, the output features in the second stage) are neural networks learned adaptively from data, with the sole requirement that the final layer of the first stage should be linear. The proposed method is shown to converge to the true causal parameter, and outperforms the recent state-of-the-art methods on challenging causal benchmarks, including settings involving high-dimensional image data.

## 1 Introduction

The goal of causal inference from observational data is to predict the effect of our actions, or *treatments*, on the *outcome* without performing interventions. Questions of interest can include *what is the effect of smoking on life expectancy?* or counterfactual questions, such as *given the observed health outcome for a smoker, how long would they have lived had they quit smoking?* Answering these questions becomes challenging when a *confounder* exists, which affects both treatment and the outcome, and causes bias in the estimation. Causal estimation requires us to correct for this confounding bias.

A popular assumption in causal inference is the *no unmeasured confounder* requirement, which means that we observe all the confounders that cause the bias in the estimation. Although a number of causal inference methods are proposed under this assumption (Hill, 2011; Shalit et al., 2017; Shi et al., 2019; Schwab et al., 2020), it rarely holds in practice. In the smoking example, the confounder can be one's genetic characteristics or social status, which are difficult to measure for both technical and ethical reasons.

To address this issue, Pearl (1995) proposed *back-door adjustment* and *front-door adjustment*, which recover the causal effect in the presence of hidden confounders using a *back-door variable* or *front-door variable*, respectively. The back-door variable is a covariate that blocks all causal effects directed from the confounder to the treatment. In health care, patients may have underlying predispositions to illness due to genetic or social factors (hidden), which cause measurable symptoms. The symptoms can be used as the back-door variable if the treatment is chosen based on these.

By contrast, a front-door variable blocks the path from treatment to outcome. In perhaps the best-known example, the amount of tar in a smoker's lungs serves as a front-door variable, since it is increased by smoking, shortens life expectancy, and has no direct link to underlying (hidden) sociological traits. Pearl (1995) showed that causal quantities can be obtained by taking the (conditional) expectation of the conditional average outcome.

While Pearl (1995) only considered the discrete case, this framework was extended to the continuous case by Singh et al. (2020), using two-stage regression (a review of this and other recent approaches

for the continuous case is given in Section 5). In the first stage, the approach regresses from the relevant covariates to the outcome of interest, expressing the function as a linear combination of non-linear feature maps. Then, in the second stage, the causal parameters are estimated by learning the (conditional) expectation of the non-linear feature map used in the first stage. Unlike competing methods (Colangelo & Lee, 2020; Kennedy et al., 2017), two-stage regression avoids fitting probability densities, which is challenging in high-dimensional settings (Wasserman, 2006, Section 6.5). Singh et al. (2020)'s method is shown to converge to the true causal parameters and exhibits better empirical performance than competing methods.

One limitation of the methods in Singh et al. (2020) is that they use fixed pre-specified feature maps from reproducing kernel Hilbert spaces, which have a limited expressive capacity when data are complex (images, text, audio). To overcome this, we propose to employ a *neural mean embedding* approach to learning task-specific adaptive feature dictionaries. At a high level, we first employ a neural network with a linear final layer in the first stage. For the second stage, we learn the (conditional) mean of the stage 1 features in the penultimate layer, again with a neural net. The approach develops the technique of Xu et al. (2021a;b) and enables the model to capture complex causal relationships for high-dimensional covariates and treatments. Neural network feature means are also used to represent (conditional) probabilities in other machine learning settings, such as representation learning (Zaheer et al., 2017) and approximate Bayesian inference (Xu et al., 2022). We derive the consistency of the method based on the Rademacher complexity, a result of which is of independent interest and may be relevant in establishing consistency for broader categories of neural mean embedding approaches, including Xu et al. (2021a;b). We empirically show that the proposed method performs better than other state-of-the-art neural causal inference methods, including those using kernel feature dictionaries.

This paper is structured as follows. In Section 2, we introduce the causal parameters we are interested in and give a detailed description of the proposed method in Section 3. The theoretical analysis is presented in Section 4, followed by a review of related work in Section 5. We demonstrate the empirical performance of the proposed method in Section 6, covering two settings: a classical back-door adjustment problem with a binary treatment, and a challenging back-door and front-door setting where the treatment consists of high-dimensional image data.

## 2    PROBLEM SETTING

In this section, we introduce the causal parameters and methods to estimate these causal methods, namely a *back-door adjustment* and *front-door adjustment*. Throughout the paper, we denote a random variable in a capital letter (e.g. $A$), the realization of this random variable in lowercase (e.g. $a$), and the set where a random variable takes values in a calligraphic letter (e.g. $\mathcal{A}$). We assume data is generated from a distribution $P$.

**Causal Parameters**    We introduce the target causal parameters using the potential outcome framework (Rubin, 2005). Let the treatment and the observed outcome be $A \in \mathcal{A}$ and $Y \in \mathcal{Y} \subseteq [-R, R]$. We denote the potential outcome given treatment $a$ as $Y^{(a)} \in \mathcal{Y}$. Here, we assume *no inference*, which means that we observe $Y = Y^{(a)}$ when $A = a$. We denote the hidden confounder as $U \in \mathcal{U}$ and assume *conditional exchangeability* $\forall a \in \mathcal{A}, \ Y^{(a)} \perp\!\!\!\perp A|U$, which means that the potential outcomes are not affected by the treatment assignment. A typical causal graph is shown in Figure 1a. We may additionally consider the observable confounder $O \in \mathcal{O}$, which is discussed in Appendix C.

A first goal of causal inference is to estimate the *Average Treatment Effect (ATE)*[1] $\theta_{\mathrm{ATE}}(a) = \mathbb{E}\left[Y^{(a)}\right]$, which is the average potential outcome of $A = a$. We also consider *Average Treatment Effect on the Treated (ATT)* $\theta_{\mathrm{ATT}}(a; a') = \mathbb{E}\left[Y^{(a)}|A = a'\right]$, which is the expected potential outcome of $A = a$ for those who received the treatment $A = a'$. Given no inference and conditional exchangeability assumptions, these causal parameters can be written in the following form.

**Proposition 1** (Rosenbaum & Rubin, 1983; Robins, 1986). *Given unobserved confounder $U$, which satisfies no inference and conditional exchangeability, we have*

$$\theta_{\mathrm{ATE}}(a) = \mathbb{E}_U\left[\mathbb{E}\left[Y|A = a, U\right]\right], \ \theta_{\mathrm{ATT}}(a; a') = \mathbb{E}_U\left[\mathbb{E}\left[Y|A = a, U\right]|A = a'\right].$$

If we observable additional confounder $O$, we may also consider *conditional average treatment effect (CATE)*: the average potential outcome for the sub-population of $O = o$, which is discussed

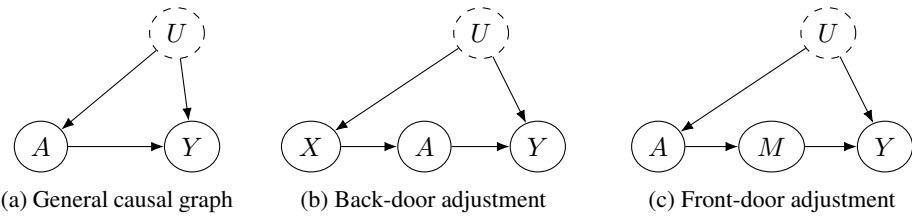

(a) General causal graph     (b) Back-door adjustment     (c) Front-door adjustment

Figure 1: Causal graphs we consider. The dotted circle means the unobservable variable.

in Appendix C. Note that since the confounder $U$ is not observed, we cannot recover these causal parameters only from $(A, Y)$.

**Back-door Adjustment** In back-door adjustment, we assume the access to the back-door variable $X \in \mathcal{X}$, which blocks all causal paths from unobserved confounder $U$ to treatment $A$. See Figure 1b for a typical causal graph. Given the back-door variable, causal parameters can be written only from observable variables $(A, Y, X)$ as follows.

**Proposition 2** (Pearl, 1995, Theorem 1)**.** *Given the back-door variable $X$, we have*

$$\theta_{\mathrm{ATE}}(a) = \mathbb{E}_X \left[ g(a, X) \right], \; \theta_{\mathrm{ATT}}(a; a') = \mathbb{E}_X \left[ g(a, X) | A = a' \right],$$

*where $g(a, x) = \mathbb{E} \left[ Y | A = a, X = x \right]$.*

By comparing Proposition 2 to Proposition 1, we can see that causal parameters can be learned by treating the back-door variable $X$ as the only "confounder", despite the presence of the additional hidden confounder $U$. Hence, we may apply any method based on the "no unobservable confounder" assumption to back-door adjustment.

**Front-door Adjustment** Another adjustment for causal estimation is *front-door adjustment*, which uses the causal mechanism to determine the causal effect. Assume we observe the front-door variable $M \in \mathcal{M}$, which blocks all causal paths from treatment $A$ to outcome $Y$, as in Figure 1c. Then, we can recover the causal parameters as follows.

**Proposition 3** (Pearl, 1995, Theorem 2)**.** *Given the front-door variable $M$, we have*

$$\theta_{\mathrm{ATE}}(a) = \mathbb{E}_{A'} \left[ \mathbb{E}_M \left[ g(A', M) | A = a \right] \right], \; \theta_{\mathrm{ATT}}(a; a') = \mathbb{E}_M \left[ g(a', M) | A = a \right],$$

*where $g(a, m) = \mathbb{E} \left[ Y | A = a, M = m \right]$ and $A' \in \mathcal{A}$ is a random variable that follows the same distribution as treatment $A$.*

Unlike the case of the back-door adjustment, we cannot naively apply methods based on the "no unmeasured confounder" assumption here, since Proposition 3 takes a different form to Proposition 1.

## 3 ALGORITHMS

In this section, we present our proposed methods. We first present the case with back-door adjustment and then move to front-door adjustment. The algorithm is summarized in Appendix A.

**Back-door adjustment** The algorithm consists of two stages; In the first stage, we learn the conditional expectation $g = \mathbb{E} \left[ Y | A = a, X = x \right]$ with a specific form. We then compute the causal parameter by estimating the expectation of the input features to $g$.

The conditional expectation $g(a, x)$ is learned by regressing $(A, X)$ to $Y$. Here, we consider a specific model $g(a, x) = \boldsymbol{w}^\top (\boldsymbol{\phi}_A(a) \otimes \boldsymbol{\phi}_X(x))$, where $\boldsymbol{\phi}_A : \mathcal{A} \to \mathbb{R}^{d_1}, \boldsymbol{\phi}_X : \mathcal{X} \to \mathbb{R}^{d_2}$ are feature maps represented by neural networks, $\boldsymbol{w} \in \mathbb{R}^{d_1 d_2}$ is a trainable weight vector, and $\otimes$ denotes a tensor product $\boldsymbol{a} \otimes \boldsymbol{b} = \mathrm{vec}(\boldsymbol{a}\boldsymbol{b}^\top)$. This tensor form used for $g(a, x)$ explicitly separates out the treatment of the features of $X$ and of $A$; in the event that $X$ is much higher dimension than $A$,

---

[1] In the binary treatment case $\mathcal{A} = \{0, 1\}$, the ATE is typically defined as the expectation of the *difference* of potential outcome $\mathbb{E}[Y^{(1)} - Y^{(0)}]$. However, we define ATE as the expectation of potential outcome $\mathbb{E}[Y^{(a)}]$, which is a primary target of interest in a continuous treatment case, also known as *dose response curve*. The same applies to the ATT as well.

then concatenating both as a single input tends to downplay the information in $A$. In addition, we can take advantage of linearity and focus on estimating the relevant (conditional) expectation as discussed later.

Given data $\{(a_i, y_i, x_i)\}_{i=1}^n \sim P$ size of $n$, the feature maps $\phi_A, \phi_X$ and the weight $\boldsymbol{w}$ can be trained by minimizing the following empirical loss:

$$\hat{\mathcal{L}}_1^{\mathcal{X}}(\boldsymbol{w}, \phi_A, \phi_X) = \frac{1}{n} \sum_{i=1}^n (y_i - \boldsymbol{w}^\top (\phi_A(a_i) \otimes \phi_X(x_i)))^2. \tag{1}$$

We may add any regularization term to this loss, such as weight decay $\lambda \|\boldsymbol{w}\|^2$. Let the minimizer of the loss $\hat{\mathcal{L}}_1^{\mathcal{X}}$ be $\hat{\boldsymbol{w}}, \hat{\phi}_A, \hat{\phi}_X = \arg\min \hat{\mathcal{L}}_1^{\mathcal{X}}$ and the learned regression function be $\hat{g}(a, x) = \hat{\boldsymbol{w}}^\top (\hat{\phi}_A(a) \otimes \hat{\phi}_X(x))$. Then, by substituting $\hat{g}$ for $g$ in Proposition 2, we have

$$\theta_{\text{ATE}}(a) \simeq \hat{\boldsymbol{w}}^\top \left( \hat{\phi}_A(a) \otimes \mathbb{E}\left[ \hat{\phi}_X(X) \right] \right), \; \theta_{\text{ATT}}(a; a') \simeq \hat{\boldsymbol{w}}^\top \left( \hat{\phi}_A(a) \otimes \mathbb{E}\left[ \hat{\phi}_X(X) \middle| A = a' \right] \right).$$

This is the advantage of assuming the specific form of $g(a, x) = \boldsymbol{w}^\top (\phi_A(a) \otimes \phi_X(x))$; From linearity, we can recover the causal parameters by estimating $\mathbb{E}[\hat{\phi}_X(X)], \mathbb{E}[\hat{\phi}_X(X)|A = a']$. Such (conditional) expectations of features are called *(conditional) mean embedding*, and thus, we name our method "*neural (conditional) mean embedding*".

We can estimate the marginal expectation $\mathbb{E}[\hat{\phi}_X(X)]$, as a simple empirical average $\mathbb{E}[\hat{\phi}_X(X)] \simeq \frac{1}{n} \sum_{i=1}^n \hat{\phi}_X(x_i)$. The conditional mean embedding $\mathbb{E}[\hat{\phi}_X(X)|A = a']$ requires more care, however: it can be learned by a technique proposed in Xu et al. (2021a), in which we train another regression function from treatment $A$ to the back-door feature $\hat{\phi}_X(X)$. Specifically, we estimate $\mathbb{E}[\hat{\phi}_X(X)|A = a']$ by $\hat{\boldsymbol{f}}_{\hat{\phi}_X}(a')$, where the regression function $\hat{\boldsymbol{f}}_{\hat{\phi}_X} : \mathcal{A} \to \mathbb{R}^{d_2}$ be given by

$$\hat{\boldsymbol{f}}_{\hat{\phi}_X} = \arg\min_{\boldsymbol{f} : \mathcal{A} \to \mathbb{R}^{d_2}} \hat{\mathcal{L}}_2^{\mathcal{X}}(\boldsymbol{f}; \hat{\phi}_X), \quad \hat{\mathcal{L}}_2^{\mathcal{X}}(\boldsymbol{f}; \phi_X) = \frac{1}{n} \sum_{i=1}^n \|\phi_X(x_i) - \boldsymbol{f}(a_i)\|^2. \tag{2}$$

Here, $\|\cdot\|$ denotes the Euclidean norm. The loss $\hat{\mathcal{L}}_2^{\mathcal{X}}$ may include the additional regularization term such as a weight decay term for parameters in $\boldsymbol{f}$. We have

$$\hat{\theta}_{\text{ATE}}(a) = \hat{\boldsymbol{w}}^\top \left( \hat{\phi}_A(a) \otimes \frac{1}{n} \sum_{i=1}^n \hat{\phi}_X(x_i) \right), \quad \hat{\theta}_{\text{ATT}}(a; a') = \hat{\boldsymbol{w}}^\top \left( \hat{\phi}_A(a) \otimes \hat{\boldsymbol{f}}_{\hat{\phi}_X}(a') \right)$$

as the final estimator for the back-door adjustment. The estimator for the ATE $\hat{\theta}_{\text{ATE}}$ is reduced to the average of the predictions $\hat{\theta}_{\text{ATE}} = \frac{1}{n} \sum_{i=1}^n \hat{g}(a, x_i)$. This coincides with other neural network causal methods (Shalit et al., 2017; Chernozhukov et al., 2022b), which do not assume $g(a, z) = \boldsymbol{w}^\top (\phi_A(a) \otimes \phi_X(x))$. As we have seen, however, this tensor product formulation is essential for estimating ATT by back-door adjustment. It will also be necessary for the front-door adjustment, as we will see next.

**Front-door adjustment** We can obtain the estimator for front-door adjustment by following the almost same procedure as the back-door adjustment. Given data $\{(a_i, y_i, m_i)\}_{i=1}^n$, we again fit the regression model $\hat{g}(a, m) = \hat{\boldsymbol{w}}^\top \left( \hat{\phi}_A(a) \otimes \hat{\phi}_M(m) \right)$ by minimizing

$$\hat{\mathcal{L}}_1^{\mathcal{M}}(\boldsymbol{w}, \phi_A, \phi_M) = \frac{1}{n} \sum_{i=1}^n (y_i - \boldsymbol{w}^\top (\phi_A(a_i) \otimes \phi_M(m_i)))^2,$$

where $\phi_M : \mathcal{M} \to \mathbb{R}^{d_2}$ is a feature map represented as the neural network. From Proposition 3, for $\boldsymbol{f}_{\hat{\phi}_M}(a) = \mathbb{E}\left[ \hat{\phi}_M(M) \middle| A = a \right]$, we have $\theta_{\text{ATE}}(a) \simeq \hat{\boldsymbol{w}}^\top \left( \mathbb{E}\left[ \hat{\phi}_A(A) \right] \otimes \boldsymbol{f}_{\hat{\phi}_M}(a) \right)$ and $\theta_{\text{ATT}}(a; a') \simeq \hat{\boldsymbol{w}}^\top \left( \hat{\phi}_A(a') \otimes \boldsymbol{f}_{\hat{\phi}_M}(a) \right)$. Again, we estimate feature embedding by empirical average for $\mathbb{E}[\hat{\phi}_A(A)]$ or solving another regression problem for $\boldsymbol{\mu}_{\hat{\phi}_M}(a)$. The final estimator for front-door adjustment is given as

$$\hat{\theta}_{\text{ATE}}(a) = \hat{\boldsymbol{w}}^\top \left( \frac{1}{n} \sum_{i=1}^n \hat{\phi}_A(a_i) \otimes \hat{\boldsymbol{f}}_{\hat{\phi}_M}(a) \right), \quad \hat{\theta}_{\text{ATT}}(a; a') = \hat{\boldsymbol{w}}^\top \left( \hat{\phi}_A(a') \otimes \hat{\boldsymbol{f}}_{\hat{\phi}_M}(a) \right),$$

where $\hat{\boldsymbol{f}}_{\hat{\phi}_M}$ is given by minimizing loss $\hat{\mathcal{L}}_2^{\mathcal{M}} = \frac{1}{n} \sum_{i=1}^n \|\phi_M(m_i) - \boldsymbol{f}(a_i)\|^2$ (with additional regularization term).

## 4 THEORETICAL ANALYSIS

In this section, we prove the consistency of the proposed method. We focus on the back-door adjustment case, since the consistency of front-door adjustment can be derived identically. The proposed method consists of two successive regression problems. In the first stage, we learn the conditional expectation $g$, and then in the second stage, we estimate the feature embeddings. First, we show each stage's consistency, then present the overall convergence rate to the causal parameter.

**Consistency for the first stage:** In this section, we consider the hypothesis space of $g$ as

$$\mathcal{H}_g = \{\boldsymbol{w}^\top(\phi_A(a) \otimes \phi_X(x)) \mid \boldsymbol{w} \in \mathbb{R}^{d_1 d_2}, \phi_A(a) \in \mathbb{R}^{d_1}, \phi_X(x) \in \mathbb{R}^{d_2},$$
$$\|\boldsymbol{w}\|_1 \le R, \max_{a \in \mathcal{A}} \|\phi_A(a)\|_\infty \le 1, \max_{x \in \mathcal{X}} \|\phi_X(x)\|_\infty \le 1\}.$$

Here, we denote $\ell_1$-norm and infinity norm of vector $\boldsymbol{b} \in \mathbb{R}^d$ as $\|\boldsymbol{b}\|_1 = \sum_{i=1}^d |b_i|$ and $\|\boldsymbol{b}\|_\infty = \max_{i \in [d]} b_i$. Note that from inequality $\|\phi_A(a) \otimes \phi_X(x)\|_\infty \le \|\phi_A(a)\|_\infty \|\phi_X(x)\|_\infty$ and Hölder's inequality, we can show that $h(a, x) \in [-R, R]$ for all $h \in \mathcal{H}_g$. First, we discuss the richness of this hypothesis space by the following theorem.

**Theorem 1.** *Let $\mathcal{A}, \mathcal{X} \subset \mathbb{R}^d$ be compact. Given sufficiently large $R, d_1, d_2$, for any continuous function $f : \mathcal{A} \times \mathcal{X} \to \mathbb{R}$ and constant $\varepsilon > 0$, there exists $h \in \mathcal{H}_g$ which satisfies $\sup_{a,x} |f(a, x) - h(a, x)| \le \varepsilon$.*

The proof uses the modified version of universal approximation theorem (Cybenko, 1989) for neural net, which will be given in Appendix B.1. Theorem 1 tells that we can approximate any continuous function $f$ with an arbitrary accuracy, which suggests the richness of our function class. Given this hypothesis space, the following lemma bounds the deviation of estimated conditional expectation $\hat{g}$ and the true one.

**Lemma 1.** *Given data $S = \{a_i, y_i, x_i\}_{i=1}^n$, let minimizer of loss $\hat{\mathcal{L}}_1^{\mathcal{X}}$ be $\hat{g} = \arg\min \hat{\mathcal{L}}_1^{\mathcal{X}}$. If the true conditional expectation $g$ is in the hypothesis space $g \in \mathcal{H}_g$, w.p. at least $1 - 2\delta$, we have*

$$\|g - \hat{g}\|_{P(A,X)} \le \sqrt{16R\hat{\mathfrak{R}}_S(\mathcal{H}_g) + 8R^2\sqrt{(\log 2/\delta)/2n}},$$

*where $\hat{\mathfrak{R}}_S(\mathcal{H}_g)$ is empirical Rademacher complexity of $\mathcal{H}_g$ given data $S$.*

The proof is given in Appendix B.3. Here, we present the empirical Rademacher complexity when we apply a feed-forward neural network for features.

**Lemma 2.** *The empirical Rademacher complexity $\hat{\mathfrak{R}}_S(\mathcal{H}_g)$ scales as $\hat{\mathfrak{R}}_S(\mathcal{H}_g) \le O(C^L/\sqrt{n})$ for some constant $C$ if we use a specific $L$-layer neural net for features $\phi_A, \phi_X$.*

See Lemma 7 in Appendix B.3 for the detailed expression of the upper bound. Note that this may be of independent interest since the similar hypothesis class is considered in Xu et al. (2021a;b), and no explicit upper bound is provided on the empirical Rademacher complexity in that work.

**Consistency for the second stage:** Next, we consider the second stage of regression. In back-door adjustment, we estimate the feature embedding $\mathbb{E}[\hat{\phi}_X(X)]$ and the conditional feature embedding $\mathbb{E}[\hat{\phi}_X(X)|A = a']$. We first state the consistency of the estimation of marginal expectation, which can be shown by Hoeffding's inequality.

**Lemma 3.** *Given data $\{x_i\}_{i=1}^n$ and feature map $\hat{\phi}_X$, w.p. at least $1 - \delta$, we have*

$$\left\|\mathbb{E}\left[\hat{\phi}_X(X)\right] - \frac{1}{n}\sum_{i=1}^n \hat{\phi}_X(x_i)\right\|_\infty \le \sqrt{\frac{2\log(2d_2/\delta)}{n}}.$$

For conditional feature embedding $\mathbb{E}[\hat{\phi}_X(X)|A = a']$, we solve the regression problem $\hat{\boldsymbol{f}}_{\hat{\phi}_X} = \arg\min_{\boldsymbol{f}} \hat{\mathcal{L}}_2^{\mathcal{X}}(\boldsymbol{f}; \hat{\phi}_X)$, the consistency of which is stated as follows.

**Lemma 4.** *Let hypothesis space $\mathcal{H}_{\boldsymbol{f}}$ be*

$$\mathcal{H}_{\boldsymbol{f}} = \{a \in \mathcal{A} \to (f_1(a), \dots, f_{d_2}(a))^\top \in [-1,1]^{d_2} \mid f_1, \dots, f_{d_2} \in \mathcal{H}_f\},$$

*where $\mathcal{H}_f$ is some hypothesis space of functions of $f : \mathcal{X} \to [-1,1]$. Let the true function be $\boldsymbol{f}_{\hat{\boldsymbol{\phi}}_X}(a) = \mathbb{E}[\hat{\boldsymbol{\phi}}_X(X)|A = a]$, and we assume $\boldsymbol{f}_{\hat{\boldsymbol{\phi}}_X} \in \mathcal{H}_{\boldsymbol{f}}$. Let $\hat{\boldsymbol{f}}_{\hat{\boldsymbol{\phi}}_X} = \arg\min_{\boldsymbol{f} \in \mathcal{H}_{\boldsymbol{f}}} \hat{\mathcal{L}}_2^{\mathcal{X}}(\boldsymbol{f}; \hat{\boldsymbol{\phi}}_X)$, given data $S = \{(a_i, x_i)\}$. Then, we have*

$$\left\| \boldsymbol{f}_{\hat{\boldsymbol{\phi}}_X}(A) - \hat{\boldsymbol{f}}_{\hat{\boldsymbol{\phi}}_X}(A) \right\|_{P(A),\infty} \leq \sqrt{16\hat{\mathfrak{R}}_S(\mathcal{H}_f) + 8\sqrt{(\log(2d_2/\delta))/2n}}$$

*w.p. at least $1 - 2\delta$, where $\|\boldsymbol{f}(A)\|_{P(A),\infty} = \max_i \|f_i\|_{P(A)}$ and $\hat{\mathfrak{R}}_S(\mathcal{H}_f)$ is the empirical Rademacher complexity of $\mathcal{H}_f$ given data $S$.*

The proof is identical to Lemma 1. We use neural network hypothesis class for $\mathcal{H}_f$ whose empirical Rademacher complexity is bounded by $O(1/\sqrt{n})$ as discussed in Proposition 5 in Appendix B.3.

**Consistency of the causal estimator**    Finally, we show that if these two estimators converge uniformly, we can recover the true causal parameters. To derive the consistency of the causal parameter, we put the following assumption on hypothesis spaces in order to guarantee that convergence in $\ell_2$-norm leads to uniform convergence.

**Assumption 1.** *For functions $h_1, h_2 \in \mathcal{H}_g$, there exists constant $c > 0$ and $\beta$ that*

$$\sup_{a \in \mathcal{A}, x \in \mathcal{X}} |h_1(a,x) - h_2(a,x)| \leq \frac{1}{c}\|h_1(A,X) - h_2(A,X)\|_{P(A,X)}^{\frac{1}{\beta}}.$$

Intuitively, this ensures that we have a non-zero probability of observing all elements in $\mathcal{A} \times \mathcal{X}$. We can see that Assumption 1 is satisfied with $\beta = 1$ and $c = \min_{(a,x) \in \mathcal{A} \times \mathcal{X}} P(A = a, X = x)$ when treatment and back-door variables are discrete. A similar intuition holds for the continuous case; in Appendix B.2, we show that Assumption 1 holds when with $\beta = \frac{2d+2}{2}$ when $\mathcal{A}, \mathcal{X}$ are $d$-dimensional intervals if the density function of $P(A, X)$ is bounded away from zero and all functions in $\mathcal{H}_g$ are Lipschitz continuous.

**Theorem 2.** *Under conditions in Lemmas 1 to 3 and Assumption 1, w.p. at least $1 - 4\delta$, we have*

$$\sup_{a \in \mathcal{A}} |\theta_{\mathrm{ATE}}(a) - \hat{\theta}_{\mathrm{ATE}}(a)| \leq O(n^{-\frac{1}{4\beta}}).$$

*If we furthermore assume that for all $\boldsymbol{f}, \tilde{\boldsymbol{f}}$,*

$$\sup_{a \in \mathcal{A}} \|\boldsymbol{f}(a) - \tilde{\boldsymbol{f}}(a)\|_\infty \leq \frac{1}{c'} \left( \max_{i \in [d_2]} \|\boldsymbol{f}(A) - \tilde{\boldsymbol{f}}(A)\|_{P(A),\infty} \right)^{\frac{1}{\beta'}},$$

*then, w.p. at least $1 - 4\delta$, we have $\sup_{a,a' \in \mathcal{A}} |\theta_{\mathrm{ATT}}(a; a') - \hat{\theta}_{\mathrm{ATT}}(a; a')| \leq O(n^{-\frac{1}{4\beta}} + n^{-\frac{1}{4\beta'}})$.*

The proof is given in Appendix B.3. This rate is slow compared to the existing work (Singh et al., 2020), which can be as fast as $O(n^{-1/4})$. However, Singh et al. (2020) assumes that the correct regression function $g$ is in a certain *reproducing kernel Hilbert space (RKHS)*, which is a stronger assumption than ours, which only assumes a Lipschitz hypothesis space. Deriving the matching minimax rates under the Lipschitz assumption remains a topic for future work.

## 5    RELATED WORK

Meanwhile learning approaches to the back-door adjustment have been extensively explored in recent work, including tree models (Hill, 2011; Athey et al., 2019), kernel models (Singh et al., 2020) and neural networks (Shi et al., 2019; Chernozhukov et al., 2022b; Shalit et al., 2017), most literature considers binary treatment cases, and few methods can be applied to continuous treatments. Schwab et al. (2020) proposed to discretize the continuous treatments and Kennedy et al. (2017); Colangelo & Lee (2020) conducted density estimation of $P(X)$ and $P(X|A)$. These are simple to implement but suffer from the curse of dimensionality (Wasserman, 2006, Section 6.5).

Recently, the automatic debiased machine learner (Auto-DML) approach (Chernozhukov et al., 2022a) has gained increasing attention, and can handle continuous treatments in the back-door adjustment. Consider a functional $m$ that maps $g$ to causal parameter $\theta = \mathbb{E}[m(g, (A, X))]$. For

the ATE case, we have $m(g, (A, X)) = g(a, X)$ since $\theta_{\text{ATE}}(a) = \mathbb{E}[g(a, X)]$. We may estimate both $g$ and the Riesz representer $\alpha$ that satisfies $\mathbb{E}[m(g, (A, X))] = \mathbb{E}[\alpha(A, X)g(A, X)]$ by the least-square regression to get the causal estimator. Although Auto-DML can learn a complex causal relationship with neural network model (Chernozhukov et al., 2022b), it requires a considerable amount of computation when the treatment is continuous, since we have to learn a different Riesz representer $\alpha$ for each treatment $a$. Furthermore, as discussed in Appendix B.4, the error bound on $\alpha$ can grow exponentially with respect to the dimension of the probability space, which may harm performance in high-dimensional settings.

Singh et al. (2020) proposed a feature embedding approach, in which feature maps are specified as the fixed feature maps in a reproducing kernel Hilbert space (RKHS). Although this strategy can be applied to a number of different causal parameters, the flexibility of the model is limited since it uses pre-specified features. Our main contribution is to generalize this feature embedding approach to adaptive features which enables us to capture more complex causal relationships. Similar techniques are used in the additional causal inference settings, such as deep feature instrumental variable method (Xu et al., 2021a) or deep proxy causal learning (Xu et al., 2021b).

By contrast with the back-door case, there is little literature that discusses non-linear front-door adjustment. The idea was originally introduced for the discrete treatment setting (Pearl, 1995) and was later discussed using the linear causal model (Pearl, 2009). To the best of our knowledge, Singh et al. (2020) is the only work that considers the nonlinear front-door adjustment, where fixed kernel feature dictionaries are used. We generalize this approach using adaptive neural feature dictionaries and obtain promising performance.

## 6 EXPERIMENTS

In this section, we evaluate the performance of the proposed method based on two scenarios. One considers the back-door adjustment methods with binary treatment based on IHDP dataset (Gross, 1993) and ACIC dataset (Shimoni et al., 2018). Another tests the performance on a high-dimensional treatment based on dSprite image dataset (Matthey et al., 2017). We first describe the training procedure we apply for our proposed method, and then report the results of each benchmark. The details of hyperparameters used in the experiment are summarized in Appendix D.

### 6.1 TRAINING PROCEDURE

During the training, we use the learning procedure proposed by Xu et al. (2021a). Let us consider the first stage regression in a back-door adjustment, in which we consider the following loss $\hat{\mathcal{L}}_1^{\mathcal{X}}$ with weight decay regularization

$$\hat{\mathcal{L}}_1^{\mathcal{X}}(\boldsymbol{w}, \boldsymbol{\phi}_A, \boldsymbol{\phi}_X) = \frac{1}{n}\sum_{i=1}^{n}(y_i - \boldsymbol{w}^\top(\boldsymbol{\phi}_A(a_i) \otimes \boldsymbol{\phi}_X(x_i)))^2 + \lambda\|\boldsymbol{w}\|^2.$$

To minimize $\hat{\mathcal{L}}_1^{\mathcal{X}}$ with respect to $(\boldsymbol{w}, \boldsymbol{\phi}_A, \boldsymbol{\phi}_X)$, we can use the closed form solution of weight $\boldsymbol{w}$. If we fix features $\boldsymbol{\phi}_A, \boldsymbol{\phi}_X$, the minimizer of $\boldsymbol{w}$ can be written

$$\hat{\boldsymbol{w}}(\boldsymbol{\phi}_A, \boldsymbol{\phi}_X) = \left(\frac{1}{n}\sum_{i=1}^{n}(\boldsymbol{\phi}_{A,X}(a_i, x_i))(\boldsymbol{\phi}_{A,X}(a_i, x_i))^\top + \lambda I\right)^{-1}\frac{1}{n}\sum_{i=1}^{n}y_i\boldsymbol{\phi}_{A,X}(a_i, x_i),$$

where $\boldsymbol{\phi}_{A,X}(a, x) = \boldsymbol{\phi}_A(a) \otimes \boldsymbol{\phi}_Z(a)$. Then, we optimize the features as $\hat{\boldsymbol{\phi}}_A, \hat{\boldsymbol{\phi}}_X = \arg\min_{\boldsymbol{\phi}_A, \boldsymbol{\phi}_X} \hat{\mathcal{L}}_1^{\mathcal{X}}(\hat{\boldsymbol{w}}(\boldsymbol{\phi}_A, \boldsymbol{\phi}_X), \boldsymbol{\phi}_A, \boldsymbol{\phi}_X)$ using Adam (Kingma & Ba, 2015). We empirically found that this stabilizes the learning and improves the performance of the proposed method.

### 6.2 BINARY TREATMENT SCENARIO

In this section, we report the performans on two classical causal datasets: IHDP dataset and ACIC dataset. The IHDP dataset is widely used to evaluate the performance of the estimators for the ATE (Shi et al., 2019; Chernozhukov et al., 2022b; Athey et al., 2019). This is a semi-synthetic dataset based on the Infant Health and Development Program (IHDP) (Gross, 1993). Following existing work, we generate 1000 sets of 747 observations of outcomes and binary treatments based

Table 1: Mean and standard error of the ATE prediction error.

|  | IHDP | ACIC |
| --- | --- | --- |
| DragonNet | $0.146 \pm 0.010$ | $0.241 \pm 0.123$ |
| RieszNet(Direct) | $0.123 \pm 0.004$ | $0.334 \pm 0.133$ |
| RieszNet(IPW) | $0.122 \pm 0.037$ | $52.73 \pm 40.71$ |
| RieszNet(DR) | $0.110 \pm 0.003$ | $1.071 \pm 0.555$ |
| RKHS Embedding | $0.166 \pm 0.003$ | $1.785 \pm 1.398$ |
| **NN Embedding (Proposed)** | $\mathbf{0.117 \pm 0.002}$ | $\mathbf{0.231 \pm 0.112}$ |

on the 25-dimensional observable confounder in the original data. The ACIC dataset is introduced in (Shi et al., 2019), which is based on linked birth and infant death data (LBIDD) (Mathews & MacDorman, 2006). This is considered a more challenging benchmark dataset than IHDP since it contains data points with extreme propensity scores (i.e. $P(A = 1|X)$ can be very close to 0 or 1). We select 101 datasets following Shi et al. (2019) and remove outliers in each dataset using the procedure described in Appendix D.

We compare our method to competing causal methods, DragonNet (Shi et al., 2019), RieszNet (Chernozhukov et al., 2022b), and RKHS Embedding (Singh et al., 2020). DragonNet is a neural causal inference method specially designed for the binary treatment, which applies the targeted regularization (van der Laan & Rubin, 2006) to ATE estimation. RieszNet implements Auto-DML with a neural network, which learns the conditional expectation $g$ and Riesz representer $\alpha$ jointly while sharing the intermediate features. Given estimated $\hat{g}, \hat{\alpha}$, it proposes three ways to calculate the causal parameter;

$$\text{Direct} : \mathbb{E}\left[m(\hat{g}, (A, X))\right], \text{IPW} : \mathbb{E}\left[Y\hat{\alpha}(A, X)\right], \text{DR} : \mathbb{E}\left[m(\hat{g}, (A, X)) + \hat{\alpha}(A, X)(Y - \hat{g}(A, X))\right],$$

where functional $m$ maps $g$ to the causal parameter (See Section 5 for the example of functional $m$). We report the performance of each estimator in RieszNet. RKHS Embedding employs the feature embedding approach with a fixed kernel feature dictionaries.

The results are summarized in Table 1. Although RieszNet(IPW) estimator performs promisingly in IHDP, the performance degenerates for the ACIC dataset, which suggests RieszNet(IPW) is prone to extreme propensity scores. This is not surprising, since the true Riesz representer in this case is $\alpha(A, X) = \frac{A}{P(A=1|X)} - \frac{1-A}{P(A=0|X)}$, which can be very large if $P(A = 1|X)$ becomes close to 0 or 1. This also harms the performance of RieszNet(DR). We can see that the proposed method outperforms all competing methods besides RieszNet(DR) in the IHDP dataset, for which the performance is comparable ($0.117 \pm 0.002$ v.s. $0.110 \pm 0.003$).

### 6.3 High-Dimensional Treatment Scenario

To test the performance of our method of causal inference in a more complex setting, we used dSprite data (Matthey et al., 2017), which is also used as the benchmark for other high-dimensional causal inference methods (Xu et al., 2021a;b). The dSprite dataset consists of images that are $64 \times 64 = 4096$-dimensional, described by five latent parameters (`shape`, `scale`, `rotation`, `posX` and `posY`). Throughout this paper, we fix (`shape`, `scale`, `rotation`) and use `posX` $\in [0, 1]$ and `posY` $\in [0, 1]$ as the latent parameters. Based on this dataset, we propose two experiments; one is ATE estimation based on the back-door adjustment, and the other is ATT estimation based on front-door adjustment.

**Back-door Adjustment** In our back-door adjustment experiment, we consider the case where the image is the treatment. Let us sample hidden confounder $U \sim \text{Unif}(0, 1)$, and consider the back-door as $(X_1, X_2) = (U \cos \theta + \varepsilon_1, U \sin \theta + \varepsilon_2)$ where $\varepsilon_1, \varepsilon_2 \sim \mathcal{N}(0, 0.09), \theta \sim \text{Unif}(0, 2\pi)$. We define treatment $A$ as the image, where the parameters are set as `posX` $= \frac{X_1+1.5}{3}$, `posY` $= \frac{X_2+1.5}{3}$. We add Gaussian noise $\mathcal{N}(0, 0.01)$ to each pixel of images. The outcome is given as follows,

$$Y = \frac{h^2(A)}{100} + 4(U - 0.5) + \varepsilon_Y, \quad h(A) = \sum_{i,j=1}^{64} \frac{(i-1)}{64} \frac{(j-1)}{64} A_{[ij]}$$

where $A_{[ij]}$ denotes the value of the pixel at $(i, j)$ and $\varepsilon_Y$ is the noise variable sampled from $\varepsilon_Y \sim \mathcal{N}(0, 0.25)$. Each dataset consists of 5000 samples of $(Y, A, X_1, X_2)$ and we consider the problem of estimating $\theta_{\text{ATE}}(a) = h^2(A)/100$. We compare the proposed method to RieszNet and RKHS Embedding, since DragonNet is designed for binary treatments and is not applicable here. We

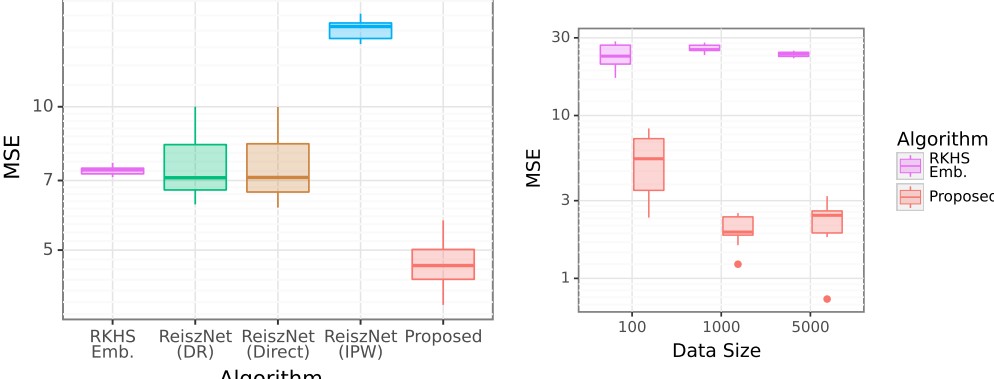

Figure 2: ATE experiment based on dSprite data

Figure 3: ATT experiment based on dSprite data

generate 10 datasets and the average of squared error $(\theta_{\text{ATE}}(a) - \hat{\theta}_{\text{ATE}}(a))^2$ at 9 test points $a$ is reported in Figure 2.

We can see that the proposed method performs best in the setting, which shows the power of the method for complex high-dimensional inputs. The RKHS Embedding method suffers from the limited flexibility of the model for the case of complex high-dimensional treatment, and performs worse than all neural methods besides RieszNet(IPW). This suggests that it is difficult to estimate Riesz representer $\alpha$ in a high-dimensional scenario, which is also suggested by the exponential growth of the error bound to the dimension as discussed in Appendix B.4. We conjecture this also harms the performance of RieszNet(Direct) and RieszNet(DR), since the models for conditional expectation $\hat{g}$ and Riesz representer $\hat{\alpha}$ share the intermediate features in the network and are jointly trained in RieszNet.

**Frontdoor Adjustment**   We use dSprite dataset to consider front-door adjustment. Again, we sample hidden confounder $U_1, U_2 \sim \text{Unif}(-1.5, 1.5)$, and we set the image to be the treatment, where the parameters are set as $\texttt{posX} = \frac{U_1 + 1.5}{3}, \texttt{posY} = \frac{U_2 + 1.5}{3}$. We add Gaussian noise $\mathcal{N}(0, 0.01)$ to each pixel of the images. We use $M = h(A) + \varepsilon_M$ as the front-door variable $M$, where $\varepsilon_M \sim \mathcal{N}(0, 0.04)$. The outcome is given as follows,

$$Y = \frac{M^2}{100} + 5(U_1 + U_2) + \varepsilon_Y, \quad \varepsilon_Y \sim \mathcal{N}(0, 0.25)$$

We consider the problem of estimating $\theta_{\text{ATT}}(a; a')$ and obtain the average squared error on 121 points of $a$ while fixing $a'$ to the image of $\texttt{posX} = 0.6, \texttt{posY} = 0.6$. We compare against RKHS Embedding, where the result is given in Figure 3. Note that RieszNet has not been developed for this setting. Again, the RKHS Embedding method suffers from the limited flexibility of the model, whereas our proposed model successfully captures the complex causal relationships.

## 7   CONCLUSION

We have proposed a novel method for back-door and front-door adjustment, based on the neural mean embedding. We established consistency of the proposed method based on a Rademacher complexity argument, which contains a new analysis of the hypothesis space with the tensor product features. Our empirical evaluation shows that the proposed method outperforms existing estimators, especially when high-dimensional image observations are involved.

As future work, it would be promising to apply a similar adaptive feature embedding approach to other causal parameters, such as *marginal average effect* $\nabla_a \theta_{\text{ATE}}(a)$ (Imbens & Newey, 2009). Furthermore, it would be interesting to consider sequential treatments, as in *dynamic treatment effect* estimation, in which the treatment may depend on the past covariates, treatments and outcomes. Recently, a kernel feature embedding approach (Singh et al., 2021) has been developed to estimate the dynamic treatment effect, and we expect that applying the neural mean embedding would benefit the performance.

ACKNOWLEDGEMENT

This work was supported by the Gatsby Charitable Foundation.

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

## A   ALGORITHM SUMMARY

Here, we provide the summary of algorithm.

---

**Algorithm 1:** Back-door Adjustment

---

**Data:** Back-door adjustment data $\{a_i, y_i, x_i\}$

1   Learn weights and features

$$\hat{\boldsymbol{w}}, \hat{\boldsymbol{\phi}}_A, \hat{\boldsymbol{\phi}}_X = \arg\min \hat{\mathcal{L}}_1^{\mathcal{X}}, \quad \hat{\mathcal{L}}_1^{\mathcal{X}} = \frac{1}{n}\sum_{i=1}^n (y_i - \boldsymbol{w}^\top(\boldsymbol{\phi}_A(a_i) \otimes \boldsymbol{\phi}_X(x_i)))^2.$$

2   Learn conditional embedding

$$\hat{\boldsymbol{f}}_{\hat{\boldsymbol{\phi}}_X} = \arg\min_{\boldsymbol{f}:\mathcal{A}\to\mathbb{R}^{d_2}} \hat{\mathcal{L}}_2^{\mathcal{X}}(\boldsymbol{f}; \hat{\boldsymbol{\phi}}_X), \quad \hat{\mathcal{L}}_2^{\mathcal{X}}(\boldsymbol{f}; \boldsymbol{\phi}_X) = \frac{1}{n}\sum_{i=1}^n \|\boldsymbol{\phi}_X(x_i) - \boldsymbol{f}(a_i)\|^2$$

3   Compute causal parameters as

$$\hat{\theta}_{\mathrm{ATE}}(a) = \hat{\boldsymbol{w}}^\top\left(\hat{\boldsymbol{\phi}}_A(a) \otimes \frac{1}{n}\sum_{i=1}^n \hat{\boldsymbol{\phi}}_X(x_i)\right)$$

$$\hat{\theta}_{\mathrm{ATT}}(a; a') = \hat{\boldsymbol{w}}^\top\left(\hat{\boldsymbol{\phi}}_A(a) \otimes \hat{\boldsymbol{f}}_{\hat{\boldsymbol{\phi}}_X}(a')\right)$$

---

---

**Algorithm 2:** Front-door Adjustment

---

**Data:** Front-door adjustment data $\{a_i, y_i, m_i\}$

1   Learn weights and features

$$\hat{\boldsymbol{w}}, \hat{\boldsymbol{\phi}}_A, \hat{\boldsymbol{\phi}}_M = \arg\min \hat{\mathcal{L}}_1^{\mathcal{M}}, \quad \hat{\mathcal{L}}_1^{\mathcal{M}} = \frac{1}{n}\sum_{i=1}^n (y_i - \boldsymbol{w}^\top(\boldsymbol{\phi}_A(a_i) \otimes \boldsymbol{\phi}_M(m_i)))^2.$$

2   Learn conditional embedding

$$\hat{\boldsymbol{f}}_{\hat{\boldsymbol{\phi}}_M} = \arg\min_{\boldsymbol{f}:\mathcal{A}\to\mathbb{R}^{d_2}} \hat{\mathcal{L}}_2^{\mathcal{M}}(\boldsymbol{f}; \hat{\boldsymbol{\phi}}_M), \quad \hat{\mathcal{L}}_2^{\mathcal{M}}(\boldsymbol{f}; \boldsymbol{\phi}_M) = \frac{1}{n}\sum_{i=1}^n \|\boldsymbol{\phi}_M(x_i) - \boldsymbol{f}(a_i)\|^2$$

3   Compute causal parameters as

$$\hat{\theta}_{\mathrm{ATE}}(a) = \hat{\boldsymbol{w}}^\top\left(\frac{1}{n}\sum_{i=1}^n \hat{\boldsymbol{\phi}}_A(a_i) \otimes \hat{\boldsymbol{f}}_{\hat{\boldsymbol{\phi}}_M}(a)\right)$$

$$\hat{\theta}_{\mathrm{ATT}}(a; a') = \hat{\boldsymbol{w}}^\top\left(\hat{\boldsymbol{\phi}}_A(a') \otimes \hat{\boldsymbol{f}}_{\hat{\boldsymbol{\phi}}_M}(a)\right)$$

---

## B   TECHNICAL DETAILS

### B.1   UNIVERSAL APPROXIMATION THEORY

In this section, we provide the proof of Theorem 1. Recall our hypothesis space is

$$\mathcal{H}_g = \{\boldsymbol{w}^\top(\boldsymbol{\phi}_A(a) \otimes \boldsymbol{\phi}_X(x)) \mid \boldsymbol{w} \in \mathbb{R}^{d_1 d_2}, \boldsymbol{\phi}_A(a) \in \mathbb{R}^{d_1}, \boldsymbol{\phi}_X(x) \in \mathbb{R}^{d_2},$$
$$\|\boldsymbol{w}\|_1 \leq R, \ \max_{a\in\mathcal{A}}\|\boldsymbol{\phi}_A(a)\|_\infty \leq 1, \ \max_{x\in\mathcal{X}}\|\boldsymbol{\phi}_X(x)\|_\infty \leq 1\}.$$

Let consider the feature be

$$\boldsymbol{\phi}_A(a) = [\sigma(s_1^\top a + \alpha_1), \ldots, \sigma(s_{d_1}^\top a + \alpha_{d_1})]^\top$$
$$\boldsymbol{\phi}_X(x) = [\sigma(t_1^\top x + \beta_1), \ldots, \sigma(t_{d_2}^\top x + \beta_{d_2})]^\top$$

where $\sigma$ is the sigmoid function and $s_i, t_i \in \mathbb{R}^D, \alpha_i, \beta_i \in \mathbb{R}$ are parameters. By considering the case of $d_1 = d_2$ and setting "non-diagonal" elements of $\boldsymbol{w}$ to zero, we can see that

$$g(a, x) = \sum_{i=1}^{d_1} w_i \sigma(s_i^\top a + \alpha_i)\sigma(t_i^\top x + \beta_i).$$

is a member of of $\mathcal{H}_g$. Next, we present the following lemma.

**Lemma 5.** *Let $\mu$ be a finite, signed regular Borel measures on $\mathcal{A} \times \mathcal{X}$. If $\sigma$ satisfies the followings:*

$$\forall s, t \in \mathbb{R}^D, \forall \alpha, \beta \in \mathbb{R}, \quad \int_{\mathcal{A}\times\mathcal{X}} \sigma(s^\top a + \alpha)\sigma(t^\top x + \beta)\mathrm{d}\mu(a, x) = 0 \Leftrightarrow \mu = 0, \qquad (3)$$

*then, given any continuous function $f : \mathcal{A} \times \mathcal{A} \to \mathbb{R}$ and $\varepsilon > 0$, there is a finite sum*

$$g(z) = \sum_{i=1}^{n} w_i \sigma(s_i^\top a + \alpha_i) \sigma(t_i^\top x + \beta_i),$$

*which satisfies*

$$\max_{a,x \in \mathcal{A} \times \mathcal{X}} |f(a,x) - g(a,x)| \le \varepsilon.$$

The proof is identical to Theorem 1 in Cybenko (1989). Now, all we have to prove is that the Sigmoid function $\sigma$ satisfies (3). This can be shown by the similar discussion as in the Lemma 1 in (Cybenko, 1989).

*Proof of Theorem 1.* Assume that

$$\forall s, t \in \mathbb{R}^D, \forall \alpha, \beta \in \mathbb{R}, \quad \int_{\mathcal{A} \times \mathcal{X}} \sigma(s^\top a + \alpha) \sigma(t^\top x + \beta) \mathrm{d}\mu(a,x) = 0$$

Then, for all $\gamma, \delta \in \mathrm{R}$, we have

$$\begin{aligned}
0 &= \lim_{\lambda_1 \to \infty} \lim_{\lambda_2 \to \infty} \int_{\mathcal{A} \times \mathcal{X}} \sigma(\lambda_1(s^\top a + \alpha) + \gamma) \sigma(\lambda_2(t^\top x + \beta) + \delta) \mathrm{d}\mu(a,x) \\
&= \int_{\mathcal{A} \times \mathcal{X}} \lim_{\lambda_1 \to \infty} \lim_{\lambda_2 \to \infty} \sigma(\lambda_1(s^\top a + \alpha) + \gamma) \sigma(\lambda_2(t^\top x + \beta) + \delta) \mathrm{d}\mu(a,x) \\
&= \int_{\mathcal{A} \times \mathcal{X}} \xi_A(a) \xi_X(x) \mathrm{d}\mu(a,x),
\end{aligned}$$

where

$$\xi_A(a) = \begin{cases} 0 & (s^\top a + \alpha < 0) \\ 1 & (s^\top a + \alpha > 0) \\ \sigma(\gamma) & (s^\top a + \alpha = 0) \end{cases}, \quad \xi_X(x) = \begin{cases} 0 & (t^\top x + \beta < 0) \\ 1 & (t^\top x + \beta > 0) \\ \sigma(\delta) & (t^\top x + \beta = 0) \end{cases}.$$

We used the Lesbegue Bounded Convergence Theorem in the second equation. From definition, we have

$$\begin{aligned}
0 &= \int_{\mathcal{A} \times \mathcal{X}} \xi_A(a) \xi_X(x) \mathrm{d}\mu(a,x) \\
&= \sigma(\gamma)\sigma(\delta)\mu(\Pi_{s,\alpha}^{\mathcal{A}} \times \Pi_{t,\beta}^{\mathcal{X}}) + \sigma(\gamma)\mu(\Pi_{s,\alpha}^{\mathcal{A}} \times H_{t,\beta}^{\mathcal{X}}) + \sigma(\delta)\mu(H_{s,\alpha}^{\mathcal{A}} \times \Pi_{t,\beta}^{\mathcal{X}}) + \mu(H_{s,\alpha}^{\mathcal{A}} \times H_{t,\beta}^{\mathcal{X}}),
\end{aligned}$$

where

$$\Pi_{s,\alpha}^{\mathcal{A}} = \{a \in \mathcal{A} | s^\top a + \alpha = 0\} \qquad \Pi_{t,\beta}^{\mathcal{X}} = \{x \in \mathcal{X} | t^\top x + \beta = 0\}$$

$$H_{s,\alpha}^{\mathcal{A}} = \{a \in \mathcal{A} | s^\top a + \alpha > 0\} \qquad H_{t,\beta}^{\mathcal{X}} = \{x \in \mathcal{X} | t^\top x + \beta > 0\}.$$

Hence for all $s, \alpha, t, \beta$, we have

$$\mu(\Pi_{s,\alpha}^{\mathcal{A}} \times \Pi_{t,\beta}^{\mathcal{X}}) = \mu(\Pi_{s,\alpha}^{\mathcal{A}} \times H_{t,\beta}^{\mathcal{X}}) = \mu(H_{s,\alpha}^{\mathcal{A}} \times \Pi_{t,\beta}^{\mathcal{X}}) = \mu(H_{s,\alpha}^{\mathcal{A}} \times H_{t,\beta}^{\mathcal{X}}) = 0.$$

Based on this, we show $\mu = 0$. Fix $s, t$ and consider functional $F(h)$ defined as

$$F(h) = \int_{\mathcal{A} \times \mathcal{X}} h(s^\top a, t^\top x) \mathrm{d}\mu(a,x),$$

where $h$ is bounded measurable function $h(u,v) : [\bar{u}, \underline{u}] \times [\bar{v}, \underline{v}] \to \mathbb{R}$, where

$$\bar{u} = \max_{a \in \mathcal{A}} s^\top a, \ \underline{u} = \min_{a \in \mathcal{A}} s^\top a, \ \bar{v} = \max_{x \in \mathcal{X}} t^\top x, \ \underline{v} = \min_{x \in \mathcal{X}} t^\top x.$$

Let indicator function $I_{(b,c] \times (d,e]}(u,v)$ defined as

$$I_{[b,c) \times [d,e)}(u,v) = \begin{cases} 1 & (u \in [b,c), v \in [d,e)) \\ 0 & \text{otherwise} \end{cases}.$$

Then, we have

$$F\left(I_{[b,\infty) \times [c,\infty)}\right) = \mu\left((\Pi_{s,-b}^{\mathcal{A}} \cup H_{s,-b}^{\mathcal{A}}) \times (\Pi_{t,-c}^{\mathcal{X}} \cup H_{t,-c}^{\mathcal{X}})\right) = 0.$$

Since

$$I_{[b,c) \times [d,e)} = I_{[b,\infty) \times [d,\infty)} - I_{[c,\infty) \times [d,\infty)} - I_{[,b\infty) \times [e,\infty)} + I_{[c,\infty) \times [e,\infty)},$$

we have $F(I_{[b,c)\times[d,e)}) = 0$ for all $b, c, d, e \in \mathbb{R}$. For linearlity, we have

$$F\left(\sum_{i=1}^{N} \eta_i I_{[b_i,c_i)\times[d_i,e_i)}\right) = 0.$$

Note that $\sum_{i=1}^{N} \eta_i I_{[b_i,c_i)\times[d_i,e_i)}$ uniformly converges to any bounded measurable function $h$ : $[\bar{u}, \underline{u}] \times [\bar{v}, \underline{v}] \to \mathbb{R}$. Hence, $F(h) = 0$. In particular, $h(u,v) = \cos(u+v), \sin(u+v)$ are bounded measurable functions, and thus,

$$\int_{\mathcal{A}\times\mathcal{X}} \exp(i(s^\top a + t^\top x))\mathrm{d}\mu(a,x)$$
$$= \int_{\mathcal{A}\times\mathcal{X}} \cos(s^\top a + t^\top x) + i\sin(s^\top a + t^\top x)\mathrm{d}\mu(a,x)$$
$$= F(\cos(u+v)) + iF(\sin(u+v)) = 0.$$

Thus, the Fourier transform of $\mu$ is 0 and so $\mu$ must be zero as well. From Lemma 5, we see Theorem 1. □

## B.2 IMPLICATION OF ASSUMPTION 1

In this section, we discuss the implication of Assumption 1, especially when the back-door and treatment variables are continuous. First, we show the upper bound of the sup norm of Lipschitz function.

**Lemma 6.** *Let $Z \in \mathcal{Z}$ be the probability variable following $P(Z)$ and $\mathcal{Z} \subset [0,1]^d$. Then, for all $L$-Lipschitz function $h$ bounded in $h(z) \in [-R, R]$, we have*

$$\max_{z\in\mathcal{Z}} |h(z)| \leq \left(\frac{4}{c}\right)^{\frac{1}{d+2}} (2R + 2\sqrt{d}L)^{\frac{d}{d+2}} \|h\|_{P(Z)}^{\frac{2}{d+2}}$$

*if the density function $f(z)$ is bounded away from zero $f(z) \geq \varepsilon > 0$.*

*Proof.* Since $\mathcal{Z}$ is compact, there exists $z^*$ such that

$$|h(z^*)| = \max_{z\in\mathcal{Z}} |h(z)|.$$

Let $M = |h(z^*)|$ and we consider the following rectangle

$$\mathfrak{B} = \left\{ z \in \mathcal{Z} \;\middle|\; \forall i \in [d] \; \max\left(0, z^*_{[i]} - \frac{M}{2R + 2\sqrt{d}L}\right) \leq z_{[i]} \leq \min\left(1, z^*_{[i]} + \frac{M}{2R + 2\sqrt{d}L}\right) \right\},$$

where $z_{[i]}$ denotes the $i$-th element of $z$. Then, from Lipschitz continuity, for all $z \in \mathfrak{B}$, we have

$$|h(z)| \geq |h(z^*)| - L\|z^* - z\|_2$$

$$= M - L\sqrt{\sum_{i=1}^{d} |z^*_{[i]} - z_{[i]}|^2}$$

$$\geq M - L\sqrt{\sum_{i=1}^{d} \left(\frac{M}{2R + 2\sqrt{d}L}\right)^2}$$

$$\geq M - L\sqrt{\sum_{i=1}^{d} \left(\frac{M}{2\sqrt{d}L}\right)^2} \geq M/2$$

Now, consider the volume of $\mathfrak{B}$. Since

$$\frac{M}{2R + 2\sqrt{d}L} \leq \frac{R}{2R + 2\sqrt{d}L} \leq \frac{R}{2R} = \frac{1}{2},$$

the events $0 \geq z^*_{[i]} - \frac{M}{2R+2\sqrt{d}L}$ and $1 \leq z^*_{[i]} + \frac{M}{2R+2\sqrt{d}L}$ do not occur simultaneously. Therefore, we have

$$\min\left(1, z^*_{[i]} + \frac{M}{2R + 2\sqrt{d}L}\right) - \max\left(0, z^*_{[i]} - \frac{M}{2R + 2\sqrt{d}L}\right) \geq \frac{M}{2R + 2\sqrt{d}L},$$

and

$$\|h\|_{P(Z)}^2 = \int_{\mathcal{Z}} |h(z)|^2 f(z) \mathrm{d}z$$

$$\geq \int_{\mathfrak{B}} |h(z)|^2 f(z) \mathrm{d}z$$

$$\geq c \left( \frac{M}{2R + 2\sqrt{d}L} \right)^d \frac{M^2}{4}.$$

Since $M = \max_{z \in \mathcal{Z}} |h(z)|$, we have

$$\max_{z \in \mathcal{Z}} |h(z)| \leq \left( \frac{4}{c} \right)^{\frac{1}{d+2}} (2R + 2\sqrt{d}L)^{\frac{d}{d+2}} \|h\|_{P(Z)}^{\frac{2}{d+2}}.$$

$\square$

By this, we can give the Assumption 1 follows for the interval probability space.

**Corollary 1.** *If $\mathcal{A} = [0,1]^{d_A}, \mathcal{X} = [0,1]^{d_X}$, and all function $h \in \mathcal{H}_g$ are $L-$Lipschitz continuous, we have*

$$\max_{a,x \in \mathcal{A} \times \mathcal{X}} |h_1(a,x) - h_2(a,x)| \leq C \|h_1 - h_2\|_{P(A,X)}^{\frac{2}{d_A + d_X + 2}},$$

*where $C = \left( \frac{4}{c} \right)^{\frac{1}{d_A + d_X + 2}} (4R + 4\sqrt{d_A + d_X}L)^{\frac{d_A + d_X}{d_A + d_X + 2}}$.*

Note that the assumption on hypothesis space is easy to satisfy since all neural network is Lipchitz function if we use the ReLU activation and regularize the operator norm of the weight in each layer.

### B.3 CONSISTENCY RESULTS

**Proof of Lemma 1** We use the following Rademacher bound to prove the consistency (Mohri et al., 2012).

**Proposition 4.** *(Mohri et al., 2012, Theorem 11.3) Let $\mathcal{X}$ be a measurable space and $\mathcal{H}$ be a family of functions mapping from $\mathcal{X}$ to $\mathcal{Y} \subseteq [-R, R]$. Given fixed dataset $S = ((y_1, x_1), (y_2, x_2), \ldots, (y_n, x_n)) \in (\mathcal{X} \times \mathcal{Y})^n$, the empirical Rademacher complexity is given by*

$$\hat{\mathfrak{R}}_S(\mathcal{H}) = \mathbb{E}_{\boldsymbol{\sigma}} \left[ \frac{1}{n} \sup_{h \in \mathcal{H}} \sum_{i=1}^n \sigma_i h(x_i) \right],$$

*where $\boldsymbol{\sigma} = (\sigma_1, \ldots, \sigma_n)$, with $\sigma_i$ independent random variables taking values in $\{-1, +1\}$ with equal probability. Then, for any $\delta > 0$, with probability at least $1 - \delta$ over the draw of an i.i.d sample $S$ of size $n$, each of following holds for all $h \in \mathcal{H}$:*

$$\mathbb{E}\left[(Y - h(X))^2\right] \leq \frac{1}{n} \sum_{i=1}^n (y_i - h(x_i))^2 + 8R\hat{\mathfrak{R}}_S(\mathcal{H}) + 4R^2 \sqrt{\frac{\log 2/\delta}{2n}},$$

$$\frac{1}{n} \sum_{i=1}^n (y_i - h(x_i))^2 \leq \mathbb{E}\left[(Y - h(X))^2\right] + 8R\hat{\mathfrak{R}}_S(\mathcal{H}) + 4R^2 \sqrt{\frac{\log 2/\delta}{2n}}.$$

Given Proposition 4, we can prove the consistency of conditional expectation.

*Proof of Lemma 1.* From Proposition 4 and $\hat{g}, g \in \mathcal{H}_g$, for the probability at least $1 - 2\delta$, we have followings.

$$\mathbb{E}\left[(Y - \hat{g}(A, X))^2\right] \leq \frac{1}{n} \sum_{i=1}^n (y_i - \hat{g}(a_i, x_i))^2 + 8R\hat{\mathfrak{R}}_S(\mathcal{H}_g) + 4R^2 \sqrt{\frac{\log 2/\delta}{2n}}$$

$$\frac{1}{n} \sum_{i=1}^n (y_i - g(a_i, x_i))^2 \leq \mathbb{E}\left[(Y - g(A, X))^2\right] + 8R\hat{\mathfrak{R}}_S(\mathcal{H}_g) + 4R^2 \sqrt{\frac{\log 2/\delta}{2n}}$$

From the minimality of $\hat{g} = \arg\min \hat{\mathcal{L}}_1^{\mathcal{X}}$, we have

$$\mathbb{E}\left[(Y - \hat{g}(A,X))^2\right] \leq \mathbb{E}\left[(Y - g(A,X))^2\right] + 16R\hat{\mathfrak{R}}_S(\mathcal{H}_g) + 8R^2\sqrt{\frac{\log 2/\delta}{2n}}$$

$$\Leftrightarrow \quad \mathbb{E}\left[(g(A,X) - \hat{g}(A,X))^2\right] \leq 16R\hat{\mathfrak{R}}_S(\mathcal{H}_g) + 8R^2\sqrt{\frac{\log 2/\delta}{2n}}.$$

Taking the square root of both sides completes the proof. $\qquad\square$

**Empirical Rademacher Complexity of $\mathcal{H}_g$**   We discuss the empirical Rademacher complexity of $\mathcal{H}_g$ when we use feed-forward neural network for features $\phi_A, \phi_X$ here. The discussion is based on a "peeling" argument proposed in Neyshabur et al. (2015).

**Proposition 5** ((Neyshabur et al., 2015), Theorem 1). *Let hypothesis space of $L$ layer neural net be $\mathcal{H}_{\mathrm{NN}}$ that*

$$\mathcal{H}_{\mathrm{NN}} = \left\{ f : \mathbb{R}^D \to \mathbb{R} \,\middle|\, f(s) = W^{(L)}\sigma\left(W^{(L-1)}\sigma(\ldots\sigma(W^{(1)}s)\right), \prod_{i=1}^{L} \|W^{(i)}\|_{p,q} \leq \gamma \right\},$$

*where $\sigma$ is ReLU function and $W^{(1)} \in \mathbb{R}^{D \times H}, W^{(L)} \in \mathbb{R}^{1 \times H}, W^{(2)}, \ldots, W^{(L-1)} \in \mathbb{R}^{H \times H}$ are weights. The norm $\|\cdot\|_{p,q}$ is matrix $L_{p,q}$-norm $\sup_{x \neq 0} \|Wx\|_q/\|x\|_p$. Then, for any $L, q \geq 1$, any $1 \leq p \leq \infty$, and any set $S = \{s_1, \ldots, s_n\}$, the empirical Rademacher complexity is bounded as*

$$\hat{\mathfrak{R}}_S(\mathcal{H}_{\mathrm{NN}}) \leq \sqrt{\frac{1}{n}\left(\gamma^2 2H^{\left[\frac{1}{p^*} - \frac{1}{q}\right]_+}\right)^{2(L-1)}(\min\{p^*, 4\log(2D)\})\max_i\|s_i\|_{p^*}}$$

*for $p^* = 1/(1 - 1/p)$ and $[x]_+ = \max\{0, x\}$.*

Given this, we can bound the empirical Rademacher complexity of $\mathcal{H}_g$ when each coordinate of features is a truncated member of $\mathcal{H}_{\mathrm{NN}}$.

**Lemma 7.** *Let $\mathcal{A}, \mathcal{X} \subset \mathbb{R}^D$ and define hypothesis set $\mathcal{H}_{\mathrm{NNFeat.}}(d)$ that*

$$\mathcal{H}_{\mathrm{NNFeat.}}(d) = \left\{\phi : \mathbb{R}^D \to \mathbb{R}^d \,\middle|\, \phi(s) = (\tilde{\sigma}(f_1(s)), \tilde{\sigma}(f_2(s)), \ldots, \tilde{\sigma}(f_d(s)))^\top, f_1, \ldots, f_d \in \mathcal{H}_{\mathrm{NN}}\right\}$$

*where $\tilde{\sigma}$ is a ramp function $\tilde{\sigma}(x) = \min(1, \max(0, x))$. Consider $\mathcal{H}_g$ that*

$$\mathcal{H}_g = \{\boldsymbol{w}^\top(\boldsymbol{\phi}_A(a) \otimes \boldsymbol{\phi}_X(x)) \mid \boldsymbol{w} \in \mathbb{R}^{d_1 d_2}, \boldsymbol{\phi}_A(a) \in \mathbb{R}^{d_1}, \boldsymbol{\phi}_X(x) \in \mathbb{R}^{d_2},$$
$$\|\boldsymbol{w}\|_1 \leq R, \ \boldsymbol{\phi}_A \in \mathcal{H}_{\mathrm{NNFeat.}}(d_1), \ \boldsymbol{\phi}_X \in \mathcal{H}_{\mathrm{NNFeat.}}(d_2)\}.$$

*Given data set $S = \{(a_1, x_1), \ldots (a_n, x_n)\}$, we have*

$$\hat{\mathfrak{R}}_S(\mathcal{H}_g) \leq 6R\sqrt{\frac{1}{n}\left(\gamma^2 2H^{\left[\frac{1}{p^*} - \frac{1}{q}\right]_+}\right)^{2(L-1)}(\min\{p^*, 4\log(2D)\})\left(\max_i\|a_i\|_{p^*} + \max_i\|x_i\|_{p^*}\right)}.$$

Note that we have

$$\max_{a \in \mathcal{A}}\|\boldsymbol{\phi}_A(a)\|_\infty \leq 1, \max_{x \in \mathcal{X}}\|\boldsymbol{\phi}_X(x)\|_\infty \leq 1$$

since we apply $\tilde{\sigma}$ in the features. The proof is given as follows.

*Proof.* Let us define the following hypothesis spaces.

$$\tilde{\mathcal{H}}_{\mathrm{NN}} = \{\tilde{\sigma} \circ f | f \in \mathcal{H}_{\mathrm{NN}}\},$$
$$\tilde{\mathcal{H}}_{\mathrm{NN}}^2 = \{\tilde{f}_1(a)\tilde{f}_2(x) | \tilde{f}_1, \tilde{f}_2 \in \tilde{\mathcal{H}}_{\mathrm{NN}}\}.$$

Then, from the definition, we have

$$\mathcal{H}_g \subset \left\{\sum_{i=1}^{d_1}\sum_{j=1}^{d_2} w_{ij}h_{ij}(a,x) \,\middle|\, \sum_{i=1}^{d_1}\sum_{j=1}^{d_2}|w_{ij}| \leq R, \forall i, j \ h_{ij} \in \tilde{\mathcal{H}}_{\mathrm{NN}}^2\right\}.$$

Since the maximum of a linear function of $\boldsymbol{w}$ over the constraint $\|\boldsymbol{w}\| \leq R$ is achieved for the values satisfying $\|\boldsymbol{w}\| = R$, we have

$$
\hat{\mathfrak{R}}_S(\mathcal{H}_g) \leq \hat{\mathfrak{R}}_S\left(\left\{\sum_{i=1}^{d_1}\sum_{j=1}^{d_2} w_{ij} h_{ij}(a,x) \,\middle|\, \sum_{i=1}^{d_1}\sum_{j=1}^{d_2} |w_{ij}| \leq R, \forall i, j \ h_{ij} \in \tilde{\mathcal{H}}_{\mathrm{NN}}^2\right\}\right)
$$

$$
= \hat{\mathfrak{R}}_S\left(\left\{\sum_{i=1}^{d_1}\sum_{j=1}^{d_2} w_{ij} h_{ij}(a,x) \,\middle|\, \sum_{i=1}^{d_1}\sum_{j=1}^{d_2} |w_{ij}| = R, \forall i, j \ h_{ij} \in \tilde{\mathcal{H}}_{\mathrm{NN}}^2\right\}\right)
$$

$$
\leq R\hat{\mathfrak{R}}_S\left(\left\{\sum_{i=1}^{d_1}\sum_{j=1}^{d_2} w_{ij} h_{ij}(a,x) \,\middle|\, \sum_{i=1}^{d_1}\sum_{j=1}^{d_2} |w_{ij}| = 1, \forall i, j \ h_{ij} \in \tilde{\mathcal{H}}_{\mathrm{NN}}^2\right\}\right)
$$

Let $\tilde{\mathcal{H}}_{\mathrm{NN}}^2 - \tilde{\mathcal{H}}_{\mathrm{NN}}^2$ be the function space defined as

$$
\tilde{\mathcal{H}}_{\mathrm{NN}}^2 - \tilde{\mathcal{H}}_{\mathrm{NN}}^2 = \left\{h_1(a,x) - h_2(a,x) \,\middle|\, h_1, h_2 \in \tilde{\mathcal{H}}_{\mathrm{NN}}^2\right\}.
$$

Since $\tilde{\mathcal{H}}_{\mathrm{NN}}^2$ contains the zero function, the final hypothesis space is the subset the convex hull of $\tilde{\mathcal{H}}_{\mathrm{NN}}^2 - \tilde{\mathcal{H}}_{\mathrm{NN}}^2$ because

$$
\sum_{i=1}^{d_1}\sum_{j=1}^{d_2} w_{ij} h_{ij}(a,x) = \sum_{w_{i,j} \geq 0} w_{ij}(h_{ij}(a,x) - 0) + \sum_{w_{i,j} < 0} |w_{ij}|(0 - h_{ij}(a,x)).
$$

Therefore, we have

$$
\hat{\mathfrak{R}}_S(\mathcal{H}_g) \leq R\hat{\mathfrak{R}}_S(\tilde{\mathcal{H}}_{\mathrm{NN}}^2 - \tilde{\mathcal{H}}_{\mathrm{NN}}^2) \leq 2R\hat{\mathfrak{R}}_S(\tilde{\mathcal{H}}_{\mathrm{NN}}^2).
$$

Now, we can bound $\hat{\mathfrak{R}}_S(\tilde{\mathcal{H}}_{\mathrm{NN}}^2)$ as

$$
\hat{\mathfrak{R}}_S(\tilde{\mathcal{H}}_{\mathrm{NN}}^2) = \hat{\mathfrak{R}}_S(\{\tilde{f}_1(a)\tilde{f}_2(x) | \tilde{f}_1, \tilde{f}_2 \in \tilde{\mathcal{H}}_{\mathrm{NN}}\})
$$

$$
= \hat{\mathfrak{R}}_S\left(\left\{\frac{1}{2}\left((\tilde{f}_1(a) + \tilde{f}_2(x))^2 - (\tilde{f}_1(a))^2 - (\tilde{f}_2(x))^2\right)\middle|\tilde{f}_1, \tilde{f}_2 \in \tilde{\mathcal{H}}_{\mathrm{NN}}\right\}\right)
$$

$$
= \frac{1}{2}\hat{\mathfrak{R}}_S\left(\left\{(\tilde{f}_1(a) + \tilde{f}_2(x))^2\middle|\tilde{f}_1, \tilde{f}_2 \in \tilde{\mathcal{H}}_{\mathrm{NN}}\right\}\right) + \frac{1}{2}\hat{\mathfrak{R}}_S\left(\left\{(\tilde{f}_1(a))^2\middle|\tilde{f}_1 \in \tilde{\mathcal{H}}_{\mathrm{NN}}\right\}\right)
$$

$$
+ \frac{1}{2}\hat{\mathfrak{R}}_S\left(\left\{(\tilde{f}_2(x))^2\middle|\tilde{f}_2 \in \tilde{\mathcal{H}}_{\mathrm{NN}}\right\}\right)
$$

$$
\leq 2\hat{\mathfrak{R}}_S\left(\left\{\tilde{f}_1(a) + \tilde{f}_2(x)\middle|\tilde{f}_1, \tilde{f}_2 \in \tilde{\mathcal{H}}_{\mathrm{NN}}\right\}\right) + \hat{\mathfrak{R}}_{S_A}(\tilde{\mathcal{H}}_{\mathrm{NN}}) + \hat{\mathfrak{R}}_{S_X}(\tilde{\mathcal{H}}_{\mathrm{NN}})
$$

$$
= 3\hat{\mathfrak{R}}_{S_A}(\tilde{\mathcal{H}}_{\mathrm{NN}}) + 3\hat{\mathfrak{R}}_{S_X}(\tilde{\mathcal{H}}_{\mathrm{NN}}),
$$

where $S_A = \{a_i\}$ and $S_X = \{x_i\}$. Here, we used Talagrand's contraction lemma (Mohri et al., 2012, Lemma 5.11) in the inequality. Again, from Talagrand's contraction lemma, we have

$$
\hat{\mathfrak{R}}_{S_A}(\tilde{\mathcal{H}}_{\mathrm{NN}}) \leq \hat{\mathfrak{R}}_{S_A}(\mathcal{H}_{\mathrm{NN}}), \ \hat{\mathfrak{R}}_{S_X}(\tilde{\mathcal{H}}_{\mathrm{NN}}) \leq \hat{\mathfrak{R}}_{S_X}(\mathcal{H}_{\mathrm{NN}}),
$$

since $\tilde{\sigma}$ is an 1-Lipchitz function.

Combining them, we have

$$
\hat{\mathfrak{R}}_S(\mathcal{H}_g) \leq 6R(\hat{\mathfrak{R}}_{S_A}(\mathcal{H}_{\mathrm{NN}}) + \hat{\mathfrak{R}}_{S_X}(\mathcal{H}_{\mathrm{NN}})).
$$

This and Proposition 5 completes the proof. $\square$

Now, we derive the final theorem to show the consistency of the method.

*Proof of Theorem 2.* From the triangular inequality, we have

$$
|\theta_{\mathrm{ATE}}(a) - \hat{\theta}_{\mathrm{ATE}}(a)| \leq |\theta - \mathbb{E}\left[\hat{g}(a,X)\right]| + \left|\hat{\theta}_{\mathrm{ATE}}(a) - \mathbb{E}\left[\hat{g}(a,X)\right]\right|
$$

For the first term of r.h.s, we have

$$
\begin{aligned}
|\theta_{\mathrm{ATE}}(a) - \mathbb{E}\left[\hat{g}(a, X)\right]| = |\mathbb{E}\left[g(a, X) - \hat{g}(a, X)\right]| \\
\leq \mathbb{E}\left[|g(a, X) - \hat{g}(a, X)|\right] \\
\leq \sup_{a \in \mathcal{A}, x \in \mathcal{X}} |g(a, x) - \hat{g}(a, x)|
\end{aligned}
$$

For the second term, we have

$$
\begin{aligned}
\left|\hat{\theta} - \int \mathbb{E}\left[\hat{g}(a, X)\right]\right| &= \left|\hat{\boldsymbol{w}}^\top \left(\hat{\boldsymbol{\phi}}_A(a) \otimes \frac{1}{n} \sum_{i=1}^{n} \hat{\boldsymbol{\phi}}_X(x_i) - \hat{\boldsymbol{\phi}}_A(a) \otimes \mathbb{E}\left[\hat{\boldsymbol{\phi}}_X(X)\right]\right)\right| \\
&\leq \|\hat{\boldsymbol{w}}\|_1 \left\|\hat{\boldsymbol{\phi}}_A(a) \otimes \frac{1}{n} \sum_{i=1}^{n} \hat{\boldsymbol{\phi}}_X(x_i) - \hat{\boldsymbol{\phi}}_A(a) \otimes \mathbb{E}\left[\hat{\boldsymbol{\phi}}_X(X)\right]\right\|_\infty \\
&\leq \|\hat{\boldsymbol{w}}\|_1 \left\|\hat{\boldsymbol{\phi}}_A(a)\right\|_\infty \left\|\frac{1}{n} \sum_{i=1}^{n} \hat{\boldsymbol{\phi}}_X(x_i) - \mathbb{E}\left[\hat{\boldsymbol{\phi}}_X(X)\right]\right\|_\infty \\
&\leq R \left\|\frac{1}{n} \sum_{i=1}^{n} \hat{\boldsymbol{\phi}}_X(x_i) - \mathbb{E}\left[\hat{\boldsymbol{\phi}}_X(X)\right]\right\|_\infty
\end{aligned}
$$

Therefore, we have

$$
|\theta_{\mathrm{ATE}}(a) - \hat{\theta}_{\mathrm{ATE}}(a)| \leq \sup_{a,x} |g(a, x) - \hat{g}(a, x)| + R \left\|\frac{1}{n} \sum_{i=1}^{n} \hat{\boldsymbol{\phi}}_X(x_i) - \mathbb{E}\left[\hat{\boldsymbol{\phi}}_X(X)\right]\right\|_\infty .
$$

Using Lemmas 1 and 3 and Assumption 1, we have

$$
\sup_{a,x} |g(a, x) - \hat{g}(a, x)| \leq \frac{1}{c} \left(16 R \hat{\mathfrak{R}}_S(\mathcal{H}_g) + 8R^2 \sqrt{\frac{\log 2/\delta}{2n}}\right)^{1/2\beta},
$$

$$
\left\|\mathbb{E}\left[\hat{\boldsymbol{\phi}}_X(X)\right] - \frac{1}{n} \sum_{i=1}^{n} \hat{\boldsymbol{\phi}}_X(x_i)\right\|_\infty \leq \sqrt{\frac{2 \log(2d_2/\delta)}{n}}
$$

with probability at least $1 - 4\delta$. Combining them and applying Lemma 7 completes the proof for ATE bound. For ATT, we can derive the followings with the same discussion

$$
|\theta_{\mathrm{ATT}}(a; a') - \hat{\theta}_{\mathrm{ATT}}(a; a')| \leq \sup_{a,x} |g(a, x) - \hat{g}(a, x)| + \sup_{a' \in \mathcal{A}} R \left\|\hat{\boldsymbol{f}}_{\hat{\boldsymbol{\phi}}_X}(a') - \mathbb{E}\left[\hat{\boldsymbol{\phi}}_X(X)|A = a'\right]\right\|_\infty .
$$

Using Lemma 4 and the assumption made in Theorem 2, we have

$$
\left\|\hat{\boldsymbol{f}}_{\hat{\boldsymbol{\phi}}_X}(a') - \mathbb{E}\left[\hat{\boldsymbol{\phi}}_X(X)|A = a'\right]\right\| \leq \frac{1}{c'} \left(16 \hat{\mathfrak{R}}_S(\mathcal{H}_f) + 8 \sqrt{\frac{\log(2d_2/\delta)}{2n}}\right)^{1/2\beta'} .
$$

If we use neural network hypothesis space $\mathcal{H}_f$ considered in Proposition 5, we can see that the ATT bound holds. $\qquad\square$

## B.4  LIMITATION OF SMOOTHNESS ASSUMPTION ON RIESZ REPRESENTER

In Chernozhukov et al. (2022b), we consider a functional $m$ such that the causal parameter $\theta$ can be written as $\theta = \mathbb{E}[m(g, (A, X))]$, where $g$ is the conditional expectation $g(a, x) = \mathbb{E}[Y|A = a, X = x]$. Then, a Riesz Representer $\alpha$, which satisfies $\mathbb{E}[m(g, (A, X))] = \mathbb{E}[\alpha(A, X)g(A, X)]$, exists as long as

$$
\mathbb{E}\left[(m^2(\alpha, (A, X))\right] \leq M \|\alpha\|_{P(A,X)}^2,
$$

for all $\alpha \in \mathcal{H}_\alpha$ and a smoothness parameter $M$. When we consider ATE $\theta_{\mathrm{ATE}}(a)$, the corresponding functional $m$ would be

$$
m(\alpha, (A, X)) = \alpha(a, X).
$$

Chernozhukov et al. (2021, Theorem 1) shows that the deviation of estimated the Riesz Representer $\hat{\alpha}$ and the true one $\alpha_0$ scales as linear to the smoothness parameter $M$.

$$
\|\hat{\alpha} - \alpha_0\|_{P(A,X)}^2 \leq O(M\delta_n + n^{-1/2}),
$$

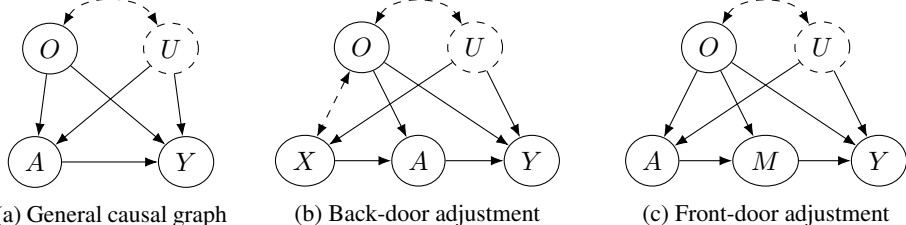

(a) General causal graph    (b) Back-door adjustment    (c) Front-door adjustment

Figure 4: Causal graph with observable confounder. The bidirectional arrows mean that we allow both directions or even a common ancestor variable.

where $\delta_n$ is the critical radius that scales

$$\delta_n = O\left(\sqrt{\frac{\log n}{n}}\right).$$

when we consider fully connected neural networks. Now, we show that the smoothness parameter $M$ can have an exponential dependency on the dimension of the space, even for simple $\alpha$. Consider $\mathcal{A} = [-1,1]^d$ and some compact space $\mathcal{X}$. We assume the uniform distribution for $P(A, X)$. Consider following $\tilde{\alpha}$

$$\tilde{\alpha}(a, x) = \max\left(1 - \sum_{i=1}^{d} 2|a_{[i]}|, 0\right),$$

where $a_{[i]}$ denotes $i$-th element of $a$, and here we consider $\tilde{\alpha}$ that does not depend on $x$. Say, we are interested in estimating $\theta_{\text{ATE}}(a)$ of $a = \mathbf{0} = [0, \ldots, 0]^\top$, for which

$$\mathbb{E}\left[(m(\tilde{\alpha})(A, X))^2\right] = \mathbb{E}\left[(\tilde{\alpha}(\mathbf{0}, X))^2\right] = 1.$$

Now consider $\mathfrak{B}$ that

$$\mathfrak{B} = \left\{a \in \mathcal{A} \;\middle|\; \forall i \in [d], -\frac{1}{2} \le a_{[i]} \le \frac{1}{2}\right\}.$$

Then, since $\tilde{\alpha}(a, x) = 0$ for all $a \notin \mathfrak{B}$, we have

$$\begin{aligned}
\|\tilde{\alpha}\|_{P(A,X)}^2 &= \int_{\mathcal{A}} |\tilde{\alpha}(A, X)|^2 \mathrm{d}P(A, X) \\
&= \int_{\mathfrak{B}} |\tilde{\alpha}(A, X)|^2 \mathrm{d}P(A, X) \\
&\le \int_{\mathfrak{B}} \mathrm{d}P(A, X) = 1/2^d.
\end{aligned}$$

We use the assumption that $P(A, X)$ is the uniform distribution to have the last equality. Hence, if $\tilde{\alpha} \in \mathcal{H}_\alpha$, the smoothness parameter $M$ must have the exponential dependency

$$M \ge 2^d.$$

## C  OBSERVABLE CONFOUNDER

In this section, we consider the case where we have the additional observable confounder, the causal graph of which is given in Figure 4.

Given the causal graph in Figure 4, ATE and ATT are defined as follows.

$$\theta_{\text{ATE}}(a) = \mathbb{E}_{U,O}\left[\mathbb{E}\left[Y|U, O, A = a\right]\right], \quad \theta_{\text{ATE}}(a; a') = \mathbb{E}_{U,O}\left[\mathbb{E}\left[Y|U, O, A = a\right]\right].$$

Furthermore, we can consider another causal parameter called *conditional average treatment effect (CATE)*, which is a conditional average of the potential outcome given $O = o$;

$$\theta_{\text{CATE}}(a; o) = \mathbb{E}\left[Y^{(a)}\middle| O = o\right].$$

Given exchangeability and no inference assumption, we have

$$\theta_{\text{CATE}}(a; o) = \mathbb{E}_{U|O=o}\left[\mathbb{E}\left[Y|U, O = o, A = a\right]\right].$$

These causal parameters can be recovered if the back-door or the front-door variable is provided as follows.

**Back-door adjustments:** First, we present the Proposition stating these causal parameters can be recovered if we are given the back-door variable $X$.

**Proposition 6** (Pearl, 1995). *Given the back-door adjustment $X$ in Figure 4b, we have*

$$\theta_{\text{ATE}}(a) = \mathbb{E}_{X,O}\left[g(a, O, X)\right],$$
$$\theta_{\text{ATT}}(a; a') = \mathbb{E}_{X,O}\left[g(a, O, X)|A = a'\right],$$
$$\theta_{\text{CATE}}(a; o) = \mathbb{E}_{X}\left[g(a, o, X)|O = o\right]$$

*where $g(a, o, x) = \mathbb{E}\left[Y|A = a, O = o, X = x\right]$.*

Now, we present the deep adaptive feature embedding approach to this. We first learn conditional expectation $\hat{g}$ as $\hat{g}(a, o, x) = \hat{\boldsymbol{w}}^\top(\hat{\boldsymbol{\phi}}_A(a) \otimes \hat{\boldsymbol{\phi}}_O(o) \otimes \hat{\boldsymbol{\phi}}_X(x))$, where

$$\hat{\boldsymbol{w}}, \hat{\boldsymbol{\phi}}_A, \hat{\boldsymbol{\phi}}_O, \hat{\boldsymbol{\phi}}_X(x) = \arg\min \frac{1}{n}\sum_{i=1}^{n}(y_i - \boldsymbol{w}^\top(\boldsymbol{\phi}_A(a_i) \otimes \boldsymbol{\phi}_O(o_i) \otimes \boldsymbol{\phi}_X(x_i)))^2 \qquad (4)$$

given data $(y_i, a_i, o_i, x_i)$. Here, $\boldsymbol{w}$ is the weight and $\boldsymbol{\phi}_A, \boldsymbol{\phi}_O, \boldsymbol{\phi}_X$ are the feature maps. From Proposition 6, we have

$$\theta_{\text{ATE}}(a) \simeq \hat{\boldsymbol{w}}^\top\left(\hat{\boldsymbol{\phi}}_A(a) \otimes \mathbb{E}_{X,O}\left[\hat{\boldsymbol{\phi}}_O(O) \otimes \hat{\boldsymbol{\phi}}_X(X)\right]\right),$$
$$\theta_{\text{ATT}}(a; a') \simeq \hat{\boldsymbol{w}}^\top\left(\hat{\boldsymbol{\phi}}_A(a) \otimes \mathbb{E}_{X,O}\left[\hat{\boldsymbol{\phi}}_O(O) \otimes \hat{\boldsymbol{\phi}}_X(X)\Big|A = a'\right]\right),$$
$$\theta_{\text{CATE}}(a; o) \simeq \hat{\boldsymbol{w}}^\top\left(\hat{\boldsymbol{\phi}}_A(a) \otimes \hat{\boldsymbol{\phi}}_O(o) \otimes \mathbb{E}\left[\hat{\boldsymbol{\phi}}_X(X)\Big|O = o\right]\right)$$

Therefore, by estimating the feature embeddings, we have

$$\hat{\theta}_{\text{ATE}}(a) = \hat{\boldsymbol{w}}^\top\left(\hat{\boldsymbol{\phi}}_A(a) \otimes \frac{1}{n}\sum_{i=1}^{n}\left(\hat{\boldsymbol{\phi}}_O(o_i) \otimes \hat{\boldsymbol{\phi}}_X(x_i)\right)\right),$$
$$\hat{\theta}_{\text{ATT}}(a; a') = \hat{\boldsymbol{w}}^\top\left(\hat{\boldsymbol{\phi}}_A(a) \otimes \hat{\boldsymbol{f}}_{\hat{\phi}_O \otimes \hat{\phi}_X}(a')\right),$$
$$\hat{\theta}_{\text{CATE}}(a; o) = \hat{\boldsymbol{w}}^\top\left(\hat{\boldsymbol{\phi}}_A(a) \otimes \hat{\boldsymbol{\phi}}_O(o) \otimes \hat{\boldsymbol{f}}_{\hat{\phi}_X}(o)\right)$$

where $\hat{\boldsymbol{f}}_{\hat{\phi}_O \otimes \hat{\phi}_X}, \hat{\boldsymbol{f}}_{\hat{\phi}_X}(o)$ are learned from

$$\hat{\boldsymbol{f}}_{\hat{\phi}_O \otimes \hat{\phi}_X} = \arg\min_{\boldsymbol{f}} \frac{1}{n}\sum_{i=1}^{n}\|\hat{\boldsymbol{\phi}}_O(o_i) \otimes \hat{\boldsymbol{\phi}}_X(x_i) - \boldsymbol{f}(a_i)\|^2$$
$$\hat{\boldsymbol{f}}_{\hat{\phi}_X} = \arg\min_{\boldsymbol{f}} \frac{1}{n}\sum_{i=1}^{n}\|\hat{\boldsymbol{\phi}}_X(x_i) - \boldsymbol{f}(o_i)\|^2.$$

**Front-door adjustment:** Given the front-door variable $M$, these causal parameters can be identified as follows.

**Proposition 7** (Pearl, 1995). *Given the front-door variable $M$ in Figure 4c, we have*

$$\theta_{\text{ATE}}(a) = \mathbb{E}_{A'}\left[\mathbb{E}_O\left[\mathbb{E}_{M|O,A=a}\left[g(A', O, M)\right]\right]\right],$$
$$\theta_{\text{ATT}}(a; a') = \mathbb{E}_O\left[\mathbb{E}_{M|O,A=a}\left[g(a', O, M)\right]\right],$$
$$\theta_{\text{CATE}}(a; o) = \mathbb{E}_{A'}\left[\mathbb{E}_{M|O=o,A=a}\left[g(A', o, M)\right]\right]$$

*where $g(a, o, m) = \mathbb{E}\left[Y|A = a, O = o, M = m\right]$ and $A'$ follows the identical distribution as $A$.*

For front-door adjustment, we learn conditional expectation $\hat{g}$ as $\hat{g}(a, o, x) = \hat{\boldsymbol{w}}^\top(\hat{\boldsymbol{\phi}}_A(a) \otimes \hat{\boldsymbol{\phi}}_O(o) \otimes \hat{\boldsymbol{\phi}}_M(m))$, where

$$\hat{\boldsymbol{w}}, \hat{\boldsymbol{\phi}}_A, \hat{\boldsymbol{\phi}}_O, \hat{\boldsymbol{\phi}}_M(m) = \arg\min \frac{1}{n}\sum_{i=1}^{n}(y_i - \boldsymbol{w}^\top(\boldsymbol{\phi}_A(a_i) \otimes \boldsymbol{\phi}_O(o_i) \otimes \boldsymbol{\phi}_M(m_i)))^2.$$

Then, from Proposition 7, we have

$$\theta_{\mathrm{ATE}}(a) \simeq \hat{\boldsymbol{w}}^\top \left( \mathbb{E}\left[ \hat{\boldsymbol{\phi}}_A(A) \right] \otimes \mathbb{E}_O \left[ \hat{\boldsymbol{\phi}}_O(O) \otimes \mathbb{E}_{M|O,A=a} \left[ \hat{\boldsymbol{\phi}}_M(M) \right] \right] \right),$$

$$\theta_{\mathrm{ATT}}(a; a') \simeq \hat{\boldsymbol{w}}^\top \left( \hat{\boldsymbol{\phi}}_A(a') \otimes \mathbb{E}_O \left[ \hat{\boldsymbol{\phi}}_O(O) \otimes \mathbb{E}_{M|O,A=a} \left[ \hat{\boldsymbol{\phi}}_M(M) \right] \right] \right),$$

$$\theta_{\mathrm{CATE}}(a; o) \simeq \hat{\boldsymbol{w}}^\top \left( \mathbb{E}\left[ \hat{\boldsymbol{\phi}}_A(A) \right] \otimes \hat{\boldsymbol{\phi}}_O(o) \otimes \mathbb{E}_{M|O=o,A=a} \left[ \hat{\boldsymbol{\phi}}_M(M) \right] \right).$$

The conditional expectation $\mathbb{E}_{M|O=o,A=a}\left[ \hat{\boldsymbol{\phi}}_M(M) \right]$ is estimated as $\mathbb{E}_{M|O=o,A=a}\left[ \hat{\boldsymbol{\phi}}_M(M) \right] = \hat{\boldsymbol{f}}_{\hat{\phi}_M}(o, a)$, where

$$\hat{\boldsymbol{f}}_{\hat{\phi}_M} = \arg\min_{\boldsymbol{f}} \frac{1}{n} \sum_{i=1}^n \| \hat{\boldsymbol{\phi}}_M(m_i) - \boldsymbol{f}(o_i, a_i) \|^2.$$

Then, by replacing the marginal expectation with the empirical average, we have

$$\hat{\theta}_{\mathrm{ATE}}(a) = \hat{\boldsymbol{w}}^\top \left( \left( \frac{1}{n} \sum_{i=1}^n \hat{\boldsymbol{\phi}}_A(a_i) \right) \otimes \frac{1}{n} \sum_{j=1}^n \left( \hat{\boldsymbol{\phi}}_O(o_j) \otimes \hat{\boldsymbol{f}}_{\hat{\phi}_M}(o_j, a) \right) \right),$$

$$\hat{\theta}_{\mathrm{ATT}}(a; a') = \hat{\boldsymbol{w}}^\top \left( \hat{\boldsymbol{\phi}}_A(a') \otimes \frac{1}{n} \sum_{i=1}^n \left( \hat{\boldsymbol{\phi}}_O(o_i) \otimes \hat{\boldsymbol{f}}_{\hat{\phi}_M}(o_i, a) \right) \right),$$

$$\hat{\theta}_{\mathrm{CATE}}(a; o) = \hat{\boldsymbol{w}}^\top \left( \left( \frac{1}{n} \sum_{i=1}^n \hat{\boldsymbol{\phi}}_A(a_i) \right) \otimes \hat{\boldsymbol{\phi}}_O(o) \otimes \hat{\boldsymbol{f}}_{\hat{\phi}_M}(o, a) \right).$$

## D  EXPERIMENT DETAILS

Here, we describe the network architecture and hyper-parameters of all experiments. Unless otherwise specified, we used Adam with learning rate = 0.001, $\beta_1 = 0.9$, $\beta_2 = 0.999$ and $\varepsilon = 10^{-8}$. For RKHS Embedding, we used Gaussian kernel for continuous variable where the bandwidth is determined by the median trick.

### D.1  BINARY TREATMENT SCENARIO

In this scenario, all treatments are binary $A \in \{0, 1\}$. In RKHS Embedding and Neural Embedding, we used the feature $\phi_A$ given as

$$\phi_A(1) = [1, 0]^\top, \phi_A(0) = [0, 1]^\top$$

in both IHDP setting and ACIC setting. This is equivalent to learn two models

$$\mathbb{E}\left[Y|X=x, A=0\right] = w_0^\top \phi_X(X), \mathbb{E}\left[Y|X=x, A=1\right] = w_1^\top \phi_X(X)$$

with shared nonlinear feature $\phi_X(X)$.

**IHDP Dataset**  We used the 1000 data used in (Chernozhukov et al., 2022b), which is publicly available at Github page of the paper. The network structure for back-door feature $\phi_X(X)$ is shown in Table 2. Note that is much smaller network than Dragonnet or Riesznet, but increasing network size did not affect the result much.

Table 2: Network structures of Neural Embedding for IHDP dataset. For the input layer, we provide the input variable. For the fully-connected layers (FC), we provide the input and output dimensions.

| Back-door feature $\phi_X(X)$ | |
| --- | --- |
| Layer | Configuration |
| 1 | Input(X) |
| 2 | FC(25, 200), ReLU |
| 3 | FC(200, 200), ReLU |

**ACIC Dataset**    We used the 101 data used in (Shi et al., 2019), which satisfies overlap assumption. (i.e. Not all data points has the extreme propensity score $P(A = 1|X)$.) We noticed that some data contains a outliers and we only consider the data points with the outcome $Y$ is in the range of

$$Y \in [Q_1(Y) - 5\text{IQR}, Q_3(Y) + 5\text{IQR}]$$

where $Q_1(Y), Q_3(Y)$ are 25%, 75%-quantile value of outcome, respectively, and $\text{IQR} = Q_3(Y) - Q_1(Y)$.

We run Dragonnet and RieszNet estimators with the same network architecture as IHDP dataset. The network structure for back-door feature $\phi_X(X)$ is shown in Table 3. Note that the same structure is used in Dragonnet and Riesznet to predict conditional expectation $\mathbb{E}[Y|X, A]$.

Table 3: Network structures of Neural Embedding for ACIC dataset. For the input layer, we provide the input variable. For the fully-connected layers (FC), we provide the input and output dimensions.

| Back-door feature $\phi_X(X)$ | |
|---|---|
| Layer | Configuration |
| 1 | Input(X) |
| 2 | FC(177, 200), ELU |
| 3 | FC(200, 200), ELU |
| 4 | FC(200, 200), ELU |
| 5 | FC(200, 100), ELU |

## D.2    HIGH-DIMENSIONAL TREATMENT SCENARIO

Here, we generate all dataset by ourselves from original dSprite dataset (Matthey et al., 2017).

**Back-door ATE estimation**    The network features for the proposed method is summarized in Table 4. The network structures for RieszNet is given in Table 5. Note that they share the similar feature extractor for images.

Table 4: Network structures of the neural embedding method in dSprite back-door adjustment experiment. For the input layer, we provide the input variable. For the fully-connected layers (FC), we provide the input and output dimensions. SN denotes Spectral Normalization (Miyato et al., 2018).

| Treatment Feature $\phi_A(A)$ | |
|---|---|
| Layer | Configuration |
| 1 | Input(A) |
| 2 | FC(4096, 1024), SN, ReLU |
| 3 | FC(1024, 512), SN, ReLU, BN |
| 4 | FC(512, 128), SN, ReLU |
| 5 | FC(128, 32), SN, BN, Tanh |

| Back-door feature $\phi_X(X)$ | |
|---|---|
| Layer | Configuration |
| 1 | Input($X$) |
| 2 | FC(2, 36), ReLU |
| 3 | FC(36, 5), ReLU |

**Front-door ATT estimation**    Here, we used the same network architecture as in the back-door adjustment summarized in Table 4.

Table 5: Network structures of RieszNet in dSprite back-door adjustment experiment. For the fully-connected layers (FC), we provide the input and output dimensions. SN denotes Spectral Normalization (Miyato et al., 2018).

| Common Feature $\phi(A, X)$ | |
|---|---|
| Layer | Configuration |
| 1 | Input($A, X$) |
| 2 | FC(4098, 1024), SN, ReLU |
| 3 | FC(1024, 512), SN, ReLU, BN |
| 4 | FC(512, 128), SN, ReLU |
| 5 | FC(128, 32), SN, BN, Tanh |

| Regressor $\hat{g}$ | |
|---|---|
| Layer | Configuration |
| 1 | Input($A, X$) |
| 2 | Common Feature $\phi(A, X)$ |
| 3 | FC(32, 32), ReLU |
| 4 | FC(32, 32), ReLU |
| 5 | FC(32, 1) |

| Riesz representer learning $\hat{\alpha}$ | |
|---|---|
| Layer | Configuration |
| 1 | Input($A, X$) |
| 2 | Common Feature $\phi(A, X)$ |
| 3 | FC(32, 1) |

