# OpenReview forum: "A Neural Mean Embedding Approach for Back-door and Front-door Adjustment"
_ICLR.cc/2023/Conference — ICLR 2023 poster_

### Official Review · Reviewer_VeMx · 2022-10-23

**Confidence:** 2
**Correctness:** 2
**Technical Novelty And Significance:** 1
**Empirical Novelty And Significance:** Not applicable
**Recommendation:** 1

**Clarity, Quality, Novelty And Reproducibility:**

The paper is written in a clear way. But I do not think the quality or novelty is sufficient for publication at ICLR.

**Strength And Weaknesses:**

Strength:

1. This paper has a thorough literature review.

2. I think the proposed method is flexible to handle multiple treatments or continuous treatments. That is an advantage compared to other methods in the literature. But on the other hand, I also have doubts / questions on the effectiveness of the method, as outlined below.

Weakness:

1. On a high level, in either back-door or front-door adjustment, how do you get the adjustment variables, X or M in the paper? Some intuitions might help the reader understand better about the problem. Also, in the experiments, you took X or M to be a noisy version of U. If the noise is small, does it imply that we have too good an adjustment variable?

2. Even though the paper claims that their method was developed in a front-door or back-door adjustment scenario, the truth is that, the authors simply assumed the existence of an adjustment variable (X or M), and developed a method that applies to the adjustment variables. In this sense, it is no different than other methodologies which make the assumption that all confounders can be observed. This is also why the proposed method can be directly compared to, e.g., dragonnet, which assumes observable confounders.

3. One big problem I can see with the proposed method is that, it does not take into account the problems with doing causal inference on observational data, namely, the distribution of X in different treatment groups could be very different. That is why various methods have been developed in the literature to account for this treatment assignment bias, e.g., doing distributional matching in Tarnet (Shalit, U., Johansson, F.D. and Sontag, D., 2017, July. Estimating individual treatment effect: generalization bounds and algorithms. In International Conference on Machine Learning (pp. 3076-3085). PMLR.)

4. The other disadvantage I can see with this two-stage regression method lies in that, the estimation of the regression models in the two stages is somehow independent. In other words, the 1st stage model cannot be informed by the 2nd stage. This is the shortcoming of a two-stage model compared to a joint model.

5. I am also curious that, when you have discrete treatments, does it pose any challenge for you to estimate the \phi_A function? How do you estimate \phi_A in this case?

6. I think you need a better way to illustrate your algorithm, either graphically or a step-by-step procedure.

7. On Page 4, bottom line, left equation, is it a typo to use f_{\phi(M)}(a)? Should you use \phi_M(m)?

8. In the experimental section, when you compared with other methods such as dragonnet, did you ensure that you used the same neural network architecture for a fair comparison? I know that the original paper of dragonnet used a very simple 2-layer neural net to estimate the conditional outcome. I am wondering if you used a more complex neural network in your method which could explain the better performance of yours. Also, dragonnet can be extended to multiple treatments case, which can also serve as a baseline in your multi-treatment experiment. You might also want to include tarnet as a baseline in the binary treatment experiment.

9. In terms of the metrics, why did you use MAE in Table 1, but MSE in Figs. 2 and 3? Also in the IHDP dataset, you did not have a front-door or back-door adjustment, right?


**Summary Of The Paper:**

This paper proposes a two-stage regression method to estimate the average treatment effects under back-door and front-door adjustment. Both stages use neural network models to learn the relevant functions. They conducted several experiments to illustrate the performance of their proposed method, using several state-of-the-art methods as baselines.

**Summary Of The Review:**

 I think the proposed method is flexible to handle multiple treatments or continuous treatments. But there are some major issues with the proposed method which I outlined in the Strength and Weakness section, e.g., it does not take into account the treatment assignment bias in the observational data. I also have doubts in the reliability of the experimental results, e.g., whether they did a fair comparison with other methods.

---

> ### Author Response · Authors · 2022-11-12
> **Comments to Reviewer VeMx (1/2)**
>
> Thank you for the detailed review and the questions raised - we clarify each point below:
>
> 1. "On a high level, in either back-door or front-door adjustment, how do you get the adjustment variables, X or M in the paper?"
>
>     The back-door variable $X$ is a covariate that blocks all causal effects directed from the confounder to the treatment. In health care, patients may have underlying predispositions to illness due to genetic or social factors (hidden), which cause measurable symptoms. These symptoms can be used as the back-door variable if the treatment is thoroughly chosen based on it.
>
>     By contrast, a front-door variable blocks the path from treatment to the outcome. In perhaps the best-known example, the amount of tar in a smoker's lungs serves as a front-door variable, since it is increased by smoking, shortens life expectancy, and has no direct link to underlying (hidden) sociological traits.
>
> 2. "Also, in the experiments, you took X or M to be a noisy version of U."
>
>     We would like to note that the front-door variable $M$ is {\em not} a noisy version of $U$ in the front-door experiment. As discussed in the experiment section, $M$ is a nonlinear function of treatment $A$, which makes $A$ and $Y$ independent given $M$. On the other hand, we did formerly set the back-door variable $X$ to be a noisy version of $U$, however in our updated experiments these now  have a nonlinear relationship. Please refer to the third point of "Comments to all reviewers".
>
> 3. "Even though the paper claims that their method was developed in a front-door or back-door adjustment scenario, the truth is that, the authors simply assumed the existence of an adjustment variable (X or M)"
>
>     Yes, we agree! The aim of our paper is to correct the confounding bias based on existing front-door and back-door variables, as originally proposed by [Rosenbaum & Rubin, 1983]. Moreover, as noted by Pearl, it is impossible in general to obtain the true causal estimation just from $(A,Y)$.
>
> 4. "In this sense, it is no different than other methodologies which make the assumption that all confounders can be observed."
>
>     In the back door case, as discussed in Proposition 2, we can see that the causal parameters can be learned by treating the back-door variable $X$ as the only "confounder" $U$ (however the graphical model permits additional unobserved confounders, as long as $X$ blocks the back door path). In the front-door adjustment setting, by contrast, no hidden confounders need be observed.
>
> 5. "One big problem"[...]"the distribution of X in different treatment groups could be very different."
>
>     We would like to note that distribution matching is not necessary. Given our assumptions, we proved that our estimator converges to the true causal parameter. It would be an interesting extension to include the discrepancy between $P(\phi(X)|T=1)$ and $P(\phi(X)|T=0)$ in the loss as in TARNet, however, and we will mention this point as future work.

---

> > ### Author Response · Authors · 2022-11-12
> > **Comments to Reviewer VeMx (2/2)**
> >
> >
> > 6. "The other disadvantage I can see with this two-stage regression method lies in that, the estimation of the regression models in the two stages is somehow independent."
> >
> >      It might be also interesting to think of end-to-end training of both stages. Nonetheless, our method achieves state-of-the-art empirical performance under the current training regime.
> >
> > 7. discrete treatments features
> >
> >      We could either partition data or (equivalently) use indicator variables. However, this scenario has been widely studied, and there are already many established methods dealing with this case, including TARNet and Dragonnet.
> >
> > 8. "I think you need a better way to illustrate your algorithm"
> >
> >      We have added an algorithm box.
> >
> > 9. "On Page 4, bottom line, left equation, is it a typo to use $f_{\phi(M)}(a)$? Should you use $\phi_M(m)?$"
> >
> >      This is not a typo; As discussed in Proposition 3, the ATE is given as $\theta_{\mathrm{ATE}}(a) = \mathbb{E}\_{A'} [\mathbb{E}\_M [g(A', M)|A=a] ]$. If we assume $g(a,m) = w^\top (\phi_A(A) \otimes \phi_M(m))$, we have
> >      $$\begin{align*}
> >        \theta_{\mathrm{ATE}}(a) = w^\top (\mathbb{E}[\phi_A(A)] \otimes \mathbb{E}[\phi_M(M)|A=a]).
> >      \end{align*}$$
> >      Hence, we use $f_{\phi(M)}(a)$ here, which is the empirical estimate of $\mathbb{E}[\phi_M(M)|A=a]$.
> >
> >
> > 10. "fair NN comparison"
> >
> >      We used the same (or smaller) network for the proposed methods in all experiments. We have added the network structure we used in the appendix. Note also that Dragonnet uses 3-layer neural nets for learning representation and additional 2-layer neural nets for estimating the conditional outcome. While Dragonnet can be extended to the multi-treatment case (as long as there are finitely many treatments), we are addressing the continuous treatment case in our dSprites experiment, to which Dragonnet does not apply.
> >      Finally, note that in the binary treatment setting, TARNet is always outperformed by Dragonnet on the datasets investigated, which is why we did not report TARNet performance.
> >
> > 11.  MAE vs MSE
> >
> >        For IHDP data, we estimate the single value ATE (i.e. $\theta_{\mathrm{ATE}}(1)- \theta_{\mathrm{ATE}}(0)$), and we take the average of the absolute error with respect to the different synthetic data. By contrary, for the dSprite data, we estimate the 9 potential outcomes and we aggregate the prediction errors by the mean squared error (i.e. 1/9 $\sum |\theta(a_i) - \hat\theta(a_i)|^2$ ). The results for different random seeds are summarized in the form of box plots. To add clarity, we change the title of Table 1 to “Mean and standard error of the ATE prediction error”.
> >
> > We hope we have addressed your concerns in the above - if so, we would be grateful if you could consider raising the review score. If any questions or concerns remain, then please let us know, and we will do our best to answer them.

---

### Official Review · Reviewer_Cbhn · 2022-10-24

**Confidence:** 4
**Correctness:** 4
**Technical Novelty And Significance:** 2
**Empirical Novelty And Significance:** 2
**Recommendation:** 6

**Clarity, Quality, Novelty And Reproducibility:**

Clarity: the paper is well written

Reproducibility: the authors provided code, and the algorithm appears easy to implement.

Novelty: I am a bit concerned about novelty, as detailed above.

**Strength And Weaknesses:**

The proposed method combines the tensor product formulation, which simplifies the estimation of ATT, and kernel learning, which leads to improved flexibility.

My main concerns are the relative lack of novelty, and the significance of the contribution:
* Broadly speaking, the idea of using learned kernels in kernelized estimators for causal effect have been well-studied (Xu et al, 2021a; 2021b).  For the specific problems of back-door and front-door adjustment in this work, the generalization of the corresponding kernel estimator (Singh et al, 2021) is straightforward, and does not appear to introduce new challenges.
* The lack of novelty will not be an issue if the benefits of the proposed method are more clearly demonstrated, but I'm not really sure if this is the case here: on the theoretical side, the authors established consistency results with worse rates than Singh et al (2021); thus, no provable benefits over fixed-form kernels were demonstrated.  On the empirical side, the empirical improvements are mainly demonstrated on the dSprite dataset, which is highly synthetic, and does not appear to relate to real-world problems well.

In aggregate, I feel there should be more demonstration of the contributions from either side, before there is a compelling case for acceptance.

**Summary Of The Paper:**

This work studies front-door and back-door adjustment using tensor product of learned kernels, generalizing Singh et al (2021) which employed fixed-form kernels.  The authors establish consistency for the estimators, and demonstrate improved performance on IHDP and a synthetic high-dimensional benchmark.

---

Post-rebuttal update: Thank you for your response. I'm raising my score given the new experiments.

**Summary Of The Review:**

Pros:

+ Proposed algorithm is a natural and intuitive extension

Cons:

- Novelty is lacking
- Neither theoretical nor practical benefits were clearly demonstrated

---

> ### Author Response · Authors · 2022-11-12
> **Comments to Reviewer Cbhn**
>
> Thank you for the review and suggestions - we address the individual points below:
>
> 1. "the generalization of the corresponding kernel estimator (Singh et al, 2021) is straightforward, and does not appear to introduce new challenges."
>
> We emphasize that learning adaptive features is more challenging than regressing on fixed features, and requires different approaches, both in theory in practice. The theoretical analysis of consistency takes a completely different form, and our proof techniques can be applied to a wide range of hypothesis spaces. The performance in experiments is significantly improved with adaptive features, making them more desirable for practitioners. Furthermore, the expressive power of the model (Refer to Common concern 1) and the explicit upper-bound on Rademacher complexity (Lemma 2)  are not considered in the previous work [Xu et al (2021a;b)], which we believe to be an important contribution.
>
>
> 2. "the authors established consistency results with worse rates than Singh et al (2021)"
>
> Consistency rates cannot be compared across the two papers, since the function classes and underlying modelling assumptions are very different. Singh et al make particular smoothness assumptions which must be satisfied for the observed rates to be attained. Our work relies on neural net features, which are arguably a richer class, but consequently make weaker smoothness assumptions suffer from slower sample rates.
>
> 3. "On the empirical side, the empirical improvements are mainly demonstrated on the dSprite dataset, which is highly synthetic and does not appear to relate to real-world problems well."
>
> To address this point, we added a new experiment based on realistic data, in which the proposed method outperforms existing methods. Please refer to the second point of "Comments to all reviewers"

---

### Official Review · Reviewer_BgB4 · 2022-10-25

**Confidence:** 3
**Correctness:** 3
**Technical Novelty And Significance:** 3
**Empirical Novelty And Significance:** 3
**Recommendation:** 6

**Clarity, Quality, Novelty And Reproducibility:**


The quality is fine aside from the lack of justification of the separability assumption made for g(A,X) and some minor/moderate deficiencies in experimental results which I outlined in the previous section.

The clarity is mostly very good aside from skipping over the separability assumption and aside from the implicit suggestion that you can always deal with hidden confounders.

The research is original to the best of my knowledge.

**Strength And Weaknesses:**

The methodology is novel to the best of my knowledge.

For the most part, I think the paper is very well written, although there are a few spots where I think the paper makes claims which are a bit too overbroad (I will elaborate shortly).

The experimental results look reasonably strong but I have a few concerns there. One concern is that ReiszNet(DR) does beat the proposed method on the health policy dataset, to a degree that may not be large but does seem to be reasonably statistically significant. I commend the authors for having the honesty to report this result, but it does weaken the case for neural mean embedding at least slightly. For the dSprite experiment, I thought the relationship between the backdoor control X and the hidden confounder U was disappointingly simple  (X1, X2 are just U1, U2 + noise). I would have thought/hoped that the authors would want to demonstrate the merits of their neural network-based technique by demonstrating the ability to capture a more complex relationship between the hidden confounder and the backdoor control.

The main overstated claim I see is that the abstract claims "Alll functions and features (and in particular, the output features in the second stage) are neural networks learned adaptively from data, with the sole requirement that the final layer of the first stage
should be linear"...but in fact, that is not the sole requirement. At the bottom of page 3, an assumption is made that g(A,X) can be represented as a linear combination of a separable function of individual neural net transformations of A and X.  I say "separable" here because that would seem to be an appropriate term, given that g has to be a linear combination of a tensor product of neural_net_1(A) times neural_net_2(X).  So A and X cannot interact arbitrarily in a nonlinear way. On page, 4, the authors refer to the advantage of this specific functional form and I don't doubt that there are advantages, but there is also the disadvantage of the possibility that some g(A,X)'s may not be implementable by the architecture, regardless of the number of hidden units used in either of the neural nets (unless I have misunderstood something). So the final layer of the first stage being linear is not the "sole requirement". I feel like the implications of this separability choice needed to be discussed/justified more. In my opinion, that would be a better use of space than the Rademacher complexity consistency/convergence stuff, which strikes me as perfectly OK but unsurprising...in general, we know that a neural net with a given number of hidden units has finite Rademacher complexity and you will converge on the correct hypothesis in the infinite data limit ...so this section seems unsurprising.

The other clarity concern I have is more minor but I would still like to see it cleaned up. There are a few spots where the writing could be taken to suggest that you can always solve the problem of having an hidden unobserved confounder with a backdoor adjustment. In fact, you have to be lucky enough to have an observed backdoor variable which closes the backdoor path between treatment and outcome. The abstract says "the goal in both cases is to recover the treatment effect without having an access to a hidden confounder".  The introduction says that "In health care, patients may have underlying predispositions to illness due to genetic or social factors (hidden), from which measurable symptoms will arise (back-door variable)".  I don't think "will arise" is appropriate here...you might be lucky enough to have access to measurable symptoms which close the backdoor paths from treatment to outcome via the hidden factors, but you also might not have access to any such measurable symptoms. Not a big problem, but some readers might get the impression that you can always deal with hidden confounders this way, and that is not true- you might not have any observed backdoor variable.

I am open to persuasion during the rebuttal phase, however.



**Summary Of The Paper:**

A method is presented for estimating causal treatment effects from observational data using neural networks for the sake of modeling flexible nonlinear relationships in the relevant conditional expectation functions. Techniques for both backdoor and frontdoor adjustments based on the directed acyclic graph causal inference theory of Pearl are presented.  The conditional expectation of outcome Y given treatment A and backdoor control X is assumed to be a linear combination of a separable tensor product of individual neural network transformations of A and X. The method has better scaling properties than competing neural-network-based causal inference methods when treatment effects are continuous and/or the problem space is high dimensional. Theoretical results based on Rademacher complexity convergence theory demonstrating the consistency of the methodology is presented. Positive experimental results on semi-synthetic health policy data and synthetic benchmark image data often used for causal inference research on images are presented.

**Summary Of The Review:**

A method is presented for estimating causal effects from observational data using neural nets to model conditional expectations. The abstract promises no assumptions or requirements for the functions to be learned aside from linearity in one output layer, but in fact the method assumes a separability between neural net transformations of the treatment and the backdoor variable. Experimental results are resonably strong but the method does not always win and one of the experiments chooses a very simple relationship between hidden confounder and backdoor variable which does not seem to take full advantage of the upside of the flexible neural net methodology.

***Update post-rebuttal ***

In light of the newly added approximation theorem and the new dSprite experiment with a more complex nonlinear control variable relationship, I have raised my score to a 6.

---

> ### Author Response · Authors · 2022-11-12
> **Comments to Reviewer BgB4**
>
> Thank you for the thorough and insightful review comments. We have updated our submission according to your suggestions:
>
> 1. "ReiszNet(DR) does beat the proposed method on the health policy dataset"
>
>     We do not believe our method (or any method) is uniformly better on every dataset, and we believe that honesty in reporting where novel methods do better and where they do worse is the best policy.
>
> 2. "relationship between the backdoor control $X$ and the hidden confounder $U$ was disappointingly simple"
>
>     We update the experimental result for dSprite which has the nonlinear relationship between $X$ and $U$. Please refer to the third point of "Comments to all reviewers ".
>
> 3. an assumption is made that g(A,X) can be represented as a linear combination of a separable function of individual neural net transformations of A and X" [...] "I feel like the implications of this separability choice needed to be discussed/justified more."
>
>     We show that our model can approximate any continuous function if $\mathcal{A} \times \mathcal{X}$ is compact. Please refer to the first point of "Comments to all reviewers ".
>
> 4. "you might be lucky enough to have access to measurable symptoms which close the backdoor paths from treatment to outcome via the hidden factors, but you also might not have access to any such measurable symptoms"
>
>     We strongly agree with this point. We reworded the claims accordingly. Please see updated submission.

---

> > ### Comment · Reviewer_BgB4 · 2022-12-05
> > **Increased score to 6**
> >
> > In light of the newly added approximation theorem and the new dSprite experiment with a more complex nonlinear control variable relationship, I have raised my score to a 6.
> >
> > I do not consider myself an expert on the interface between ML and causal inference, though, so I do not have high confidence in my score.

---

### Official Review · Reviewer_jNzo · 2022-10-28

**Confidence:** 2
**Correctness:** 4
**Technical Novelty And Significance:** 3
**Empirical Novelty And Significance:** 3
**Recommendation:** 8

**Clarity, Quality, Novelty And Reproducibility:**

The paper is well-written and follows naturally. The idea and statement are clearly stated. The proposed method is novel and straightforward. The provided theoretic analysis is interesting.

**Strength And Weaknesses:**

Strength:
1) The authors focus on the important problem, treatment effect, in the causal inference community. They leverage the approximate power from the neural networks to focus on back-door adjustment and front-door adjustment.
2) The idea is straightforward and intuitive. The theoretic analysis provides insights into the proposed method.
3) The experiment results are promising.

Weaknesses:
1) It seems that it assumes the best approximation function exists in the hypothesis function space. However, what if that is not the case?
2) It is a bit unclear to me why the function form of g(a,x) is designed in the way stated in the paper. Maybe more explanation will be better.

Question:
1) Could you please explain a bit more about why you design the L2 loss function in Equation 2?


**Summary Of The Paper:**

The paper targets continuous intervention and counterfactual problem which is usually handled by back-door and front-door adjustment in causal inference. In particular, the authors aim to improve two-stage regression limited by fixed pre-specified feature maps. To improve, the authors propose employing two neural networks to extract adaptive features and estimate the mean embedding, respectively. The loss function is the regression loss with regularization. More importantly, the authors provide with theoretical analysis of the consistency of the proposed method using Rademacher complexity, which bounds the estimated ATE, assuming all the functions in the hypothesis space are Lipschitz continuous. Lastly, the authors demonstrate the proposed method on two popular causal data sets to support their claim.



**Summary Of The Review:**

The paper proposes to train two neural networks for estimating treatment effect, an important question in causal inference. The two neural networks are trained based on regression objectives, which can be easily reproduced. The authors then provide with the theoretic analysis to bound the estimated treatment effect for intervention and counterfactual setting. The experimental results look promising and intriguing.

---

> ### Author Response · Authors · 2022-11-12
> **Comments to Reviewer jNzo**
>
> We are glad to hear you like the paper. Let us clarify the point you made:
>
> 1. It seems that it assumes the best approximation function exists in the hypothesis function space. However, what if that is not the case?
>
>     If the true conditional expectation does not exist in the hypothesis space, the causal parameter would be biased. However, the bias will be bounded in terms of the degree of misspecification (i.e. the smallest regression error achievable in the hypothesis space). Given that our hypothesis class can approximate any continuous function on compact domains, this bias should be small once sufficient training data is observed. Please refer to the first point of "Comments to all reviewers".
>
> 2. It is a bit unclear to me why the function form of g(a,x) is designed in the way stated in the paper. Maybe more explanation will be better.
>
>       This form enables the model to focus on the covariate features while ignoring the treatment features in the second stage regression.
>     The tensor form used for $g(a,x)$ explicitly separates out the treatment of the features of $X$ and of $A$; in the event that $X$ is much higher dimension than $A$, then concatenating both as a single input tends to downplay the information in $A$. In addition, if we use $g(a,x) = w^\top (\phi_A(a) \otimes \phi_X(x))$, we can take advantage of linearity and have $\mathbb{E}[g(a, X)|A=a'] =  w^\top (\phi_A(A) \otimes \mathbb{E}[\phi_X(X)|A=a'])$. Now, we can focus on learning $\mathbb{E}[\phi_X(X)|A]$ rather than learning $\mathbb{E}[g(a, X)|A=a']$. We will include this discussion in the final version.
>
> 3. Could you please explain a bit more about why you design the L2 loss function in Equation 2?
>
>     It is because the conditional expectation of feature  $\mathbb{E}[\phi_X(X)|A]$ is characterized as the minimizer of the L2 loss. Indeed,
>     $$\begin{align*}
>         \mathbb{E}[\|\phi_X(X) - f(A)\|^2] &= \mathbb{E}[\|(\phi_X(X) -  \mathbb{E}[\phi_X(X)|A] +  \mathbb{E}[\phi_X(X)|A] - f(A)\|^2]\\\\
>         &= \mathbb{E}[\|(\phi_X(X) -  \mathbb{E}[\phi_X(X)|A]\|^2] + \mathbb{E}[\| \mathbb{E}[\phi_X(X)|A] - f(A)\|^2],
>     \end{align*}$$
>     and we can see that $f(A) = \mathbb{E}[\phi_X(X)|A]$ minimizes this L2 loss. We empirically minimize this to obtain the estimation of conditional expectation $ \mathbb{E}[\phi_X(X)|A]$.

---

### Author Response · Authors · 2022-11-12
**Comments to all reviewers**

We thank all reviewers for their insightful comments. We have updated our submission to address the suggestions made by the reviewers, in which new material is shown in red. Please see the updated document for details.


1. Expressive power of model form $g(a,x) = w^\top (\phi_A(A) \otimes \phi_X(X))$

      One common concern is expressive power. The flexibility of the model is suggested by the strong empirical results, in which the model successfully learns the causal parameters when the treatment is the image. The same holds when it is applied to other causal settings [Xu et al (2021a;b)].

    Moreover, to address this theoretically, we added a new theorem, which states that given sufficiently large dimensions, our model can approximate any continuous function in a compact domain with arbitrary accuracy. We show this by extending the result known as the universal approximation theorem [Cybenko, 1989]. This suggests that assuming this specific form retains strong expressive power.

2. New experiment with binary treatment

    Some reviewers pointed out that the proposed method only performs better in the settings which we proposed. To address this point, we added a new experiment based on the ACIC dataset. This is introduced by [Shi et al. 2019], which is based on the real linked birth and infant death data (LBIDD). This benchmark is considered to be harder than the IHDP dataset since it includes more various data generation processes and contains data points with extreme propensity scores (i.e. $P(T=1|X)$ can be close to 0 or 1). We showed that the proposed method outperforms existing methods in this ACIC experiment.

3. New back-door adjustment experiment with dSprite dataset

    Some reviewers pointed out that the back-door $X$ in the dSprite experiment is too simple since it is just a noisy version of confounder $U$. To address this point, we have updated the experiment to include a nonlinear relationship between the back-door $X$ and the confounder $U$. Specifically, we sample the hidden confounder $U \sim \mathrm{Unif}(0, 1)$ and use the back-door as $X_1 = U \cos\theta + \varepsilon_1, X_2 = U \sin\theta + \varepsilon_2$ where $\theta \sim \mathrm{Unif}(0, 2\pi)$.

---

### Decision · Program_Chairs · 2023-01-20

**Decision:**

Accept: poster

**Justification For Why Not Higher Score:**

The paper makes some novel and exciting contributions but the approach empirical performance is somewhat underwhelming in certain cases.

**Justification For Why Not Lower Score:**

The paper proposes a novel and simple approach that is backed by a very nice theoretical analysis which can benefit several works beyond this one. The empirical evaluation is extensive and the authors have done their best to address the reviewers concerns.

**Metareview: Summary, Strengths And Weaknesses:**

The paper studies causal inference in the presence of hidden confounders. Using neural mean embedding, a novel approach is proposed for back-door and front-door adjustment. Consistency results are provided fro the proposed approach. These are complemented by empirical studies under binary and high-dimensional treatment scenarios.

The paper makes several interesting contributions and some results are pertinent to other studies. For instance, the Rademacher complexity bounds should directly benefit related work. The reviewers and AC carefully examined the author feedback and revisions of the manuscript. These satisfactorily and convincingly address the reviewers concerns and greatly improve the manuscript.

The authors have outlined some exciting direction for future work and we strongly encourage them to follow up on these!



**Note From Pc:**

if the above contains the word "oral" or "spotlight" please see: "oral" presentation means -> notable-top-5% and "spotlight" means -> notable-top-25%. As stated in our emails, we are disassociating presentation type from AC recommendations